# Multiple mechanisms contribute to fluorometry signals from the voltage-gated proton channel

Ferenc Papp[1✉], Gilman E. S. Toombes [2], Zoltán Pethő [1,3], Adrienn Bagosi[1], Adam Feher[1], János Almássy[4], Jesús Borrego[1], Ákos Kuki[5], Sándor Kéki[5], Gyorgy Panyi[1] & Zoltan Varga[1]

Voltage-clamp fluorometry (VCF) supplies information about the conformational changes of voltage-gated proteins. Changes in the fluorescence intensity of the dye attached to a part of the protein that undergoes a conformational rearrangement upon the alteration of the membrane potential by electrodes constitute the signal. The VCF signal is generated by quenching and dequenching of the fluorescence as the dye traverses various local environments. Here we studied the VCF signal generation, using the Hv1 voltage-gated proton channel as a tool, which shares a similar voltage-sensor structure with voltage-gated ion channels but lacks an ion-conducting pore. Using mutagenesis and lipids added to the extracellular solution we found that the signal is generated by the combined effects of lipids during movement of the dye relative to the plane of the membrane and by quenching amino acids. Our 3-state model recapitulates the VCF signals of the various mutants and is compatible with the accepted model of two major voltage-sensor movements.

---

[1] Department of Biophysics and Cell Biology, Faculty of Medicine, University of Debrecen, Egyetem ter 1, Debrecen H-4032, Hungary. [2] Molecular Physiology and Biophysics Section, Porter Neuroscience Research Center, National Institute of Neurological Disorders and Stroke, National Institutes of Health, 35 Convent Dr., MSC 3701, Bethesda, MD 20892-3701, USA. [3] Institut für Physiologie II, Robert-Koch-Str. 27b, 48149 Münster, Germany. [4] Department of Physiology, Faculty of Medicine, University of Debrecen, Egyetem ter 1, Debrecen H-4032, Hungary. [5] Department of Applied Chemistry, University of Debrecen, Egyetem ter 1, Debrecen H-4032, Hungary. ✉email: papp.ferenc@med.unideb.hu

**Generation of the VCF signal**. Voltage-Clamp Fluorometry (VCF) is an electrophysiological technique that simultaneously provides data about the ionic current flowing through the ion channel and the concurrent conformational change occurring in the protein via fluorescence[1,2]. VCF is performed on *Xenopus laevis* oocytes expressing the target protein, which is usually voltage sensitive. Before expression, an amino acid is replaced by a cysteine in the part of the protein suspected to undergo a conformational change in response to a change in membrane voltage. Following expression, the cysteine is labeled with a fluorescent dye, which is sensitive to its local environment. This fluorophore can then provide the VCF signal, by which me mean the change in the emitted fluorescence intensity divided by the holding fluorescence intensity, i.e. $\Delta F/F_h$ in response to conformational changes induced by membrane depolarization.

There have been various explanations for the VCF signal generation since the introduction of the VCF technique: (I) Quenching by nearby segments of the protein containing quenching amino acids close to the fluorophore. This is exemplified by a tryptophan (Trp) inserted into the tandem *Ciona intestinalis* Hv1 proton channel (Ci-Hv1)[3], into TMEM266, a voltage-sensitive protein of currently unknown function[4], or into the human BK channel[5]. (II) Movement of the fluorophore between less polar and aqueous environments[1,2,6], i.e., from an area surrounded by the membrane to an area more exposed to the extracellular solution, respectively. (III) The Stark or electrochromic effect[7–9], in which the local electric field directly alters the energy levels of the dye, modifying and perturbing the ground or excited energy states of the fluorophore.

The above three experimentally supported explanations are the most accepted for the formation of the VCF signal. However, how much each of these factors contributes to the formation of the VCF signal in a particular protein has not been elucidated yet. Knowledge of the mechanism of the VCF signal generation yields additional information about the conformational change of the protein and can help to better understand the previously published VCF results for the different proteins.

**The voltage-gated proton channel Hv1**. Proteins known to have voltage-sensing ability were traditionally voltage-gated ion channels. By now, three proteins have been discovered, which contain voltage-sensing domains (VSD), but lack the pore domain (PD) through the cell membrane for the passage of ions: the voltage-sensitive phosphatase (VSP)[10], TMEM266[4] and Hv1[11,12]. The Hv1 protein could be subsequently associated with the previously well-known proton current (first described almost four decades ago[13]) and then was named Hv1, voltage-gated proton channel. By now it is known that Hv1 is expressed in multiple cell types in the human body, such as in different immune cells[14,15], lung epithelial cells[16], sperm[17], pancreatic β-cells[18,19] and stem cells[20]. The presence and possible functional relevance of Hv1 have been described in cancer cells as well[21–24]. The physiological role of Hv1 is best understood in macrophages, where the NADPH oxidase complex generates reactive oxygen species (ROS) to kill bacteria. This process is electrogenic, depolarizes the membrane and acidifies the cytosol. The activity of Hv1 is a counteraction, expelling protons and repolarizing the cell membrane[14,25,26]. Hv1 function in sperm motility has also been well characterized[17,27,28].

Hv1 functions as a dimer in nature[29–31]. Based on a homology model[32] and the X-ray crystal structure[33], Hv1 can be visualized as two VSDs next to each other, which are connected and coupled together, with strong cooperativity between the two subunits during gating[34]. Although monomeric Hv1 is also capable of transmitting a proton current, the kinetics of this current differs from that of the dimer. Cooperativity means that in the dimeric

Hv1 channel, the S4 of both subunits must move to activate the two proton permeation pathways. In contrast, if Hv1 subunits are prevented from dimerizing, the movement of a single S4 is sufficient to activate the proton permeation pathway in a subunit[34]. The Isacoff and Larsson labs have successfully applied the VCF technique to study Hv1, collecting information on the gating and the conformational changes. Both groups recorded complex VCF signals with multiple components indicating the presence of at least two distinct conformational changes[35,36]. The model developed by the Isacoff lab suggested two conformational changes: a voltage-sensing transition, during which the two VSDs tend to move closely together due to strong positive cooperativity and a subsequent opening transition. According to the model of the Larsson lab, in response to a depolarizing pulse, the two S4 segments in a dimeric Hv1 channel move outward independently of each other and cause the decrease of fluorescence intensity. Subsequently, there is a second concerted conformational change of both subunits that opens the two permeation pathways and increases the fluorescence to an intermediate level producing the biphasic VCF signal. Direct measurement of S4 motion via gating charge was published in 2018[37,38], which provided strong proof that the voltage sensing mechanism of Hv1 is the same as in other voltage-gated ion channels. The study concluded that the movement of the voltage sensor must proceed through at least five states to account for the experimental data satisfactorily, and thus gave a more detailed mechanism of gating.

In this work, we have studied the origin of the VCF signal in Ci-Hv1 to understand the mechanism behind signal generation (specifically for Hv1 and for the VCF technique in general) and to draw conclusions about conformational changes. To elucidate the origins of the VCF signal, we systematically analyzed the plausible causes detailed in the section above: (I) Amino acids assumed to contribute to signal formation were mutated to assess the role of fluorescence quenching; (II) We added lipid (Phosphatidylcholine (PC)) vesicles to the external solution to investigate the contribution of lipid-aqueous transitions in signal generation. (III) We compared our VCF signal parameters to the electrochromic one.

## Results

**Effects of His to Ala mutations on the VCF signal**. In Ci-Hv1 labeled with TAMRA-MTS on E241C, upon depolarization VCF signals showed a fast quenching component followed by a slower dequenching component. The signal had similar components upon repolarization, but with greater amplitude and faster kinetics resulting in a shape previously referred to as the "hook" (Fig. 1a), which agrees with results published earlier[35]. Due to the complexity of the VCF signal, the magnitude of various components can be measured (Fig. 1a): $\Delta F_{peak}$ is the negative peak fluorescence change during the voltage step, $\Delta F_{ss}$ (steady state) is the change in fluorescence at the end of the voltage step, $\Delta F_{tail}$ is the maximum change in fluorescence during repolarization after the voltage step and $\Delta F_{hook} = \Delta F_{tail} - \Delta F_{ss}$. For consistent comparisons, we used $\Delta F_{tail}$, since for most constructs this component was the largest. In TMEM266, the VCF signal showed similar complexity and a His residue in the extracellular part was shown to play a role in VCF signal generation[4]. Histidine was also found to be one of the four residues able to quench the fluorescence of Alexa488 in solution along with Trp, Tyr and Met[39]. We have found using spectrofluorometry that His can quench TAMRA-MTS as well (Supplementary Fig. 1b and[4]), thus supporting the theory that quenching by certain amino acids may contribute to $\Delta F$ generation. We examined the sequence of Hv1 looking for quenching residues in the extracellular region that

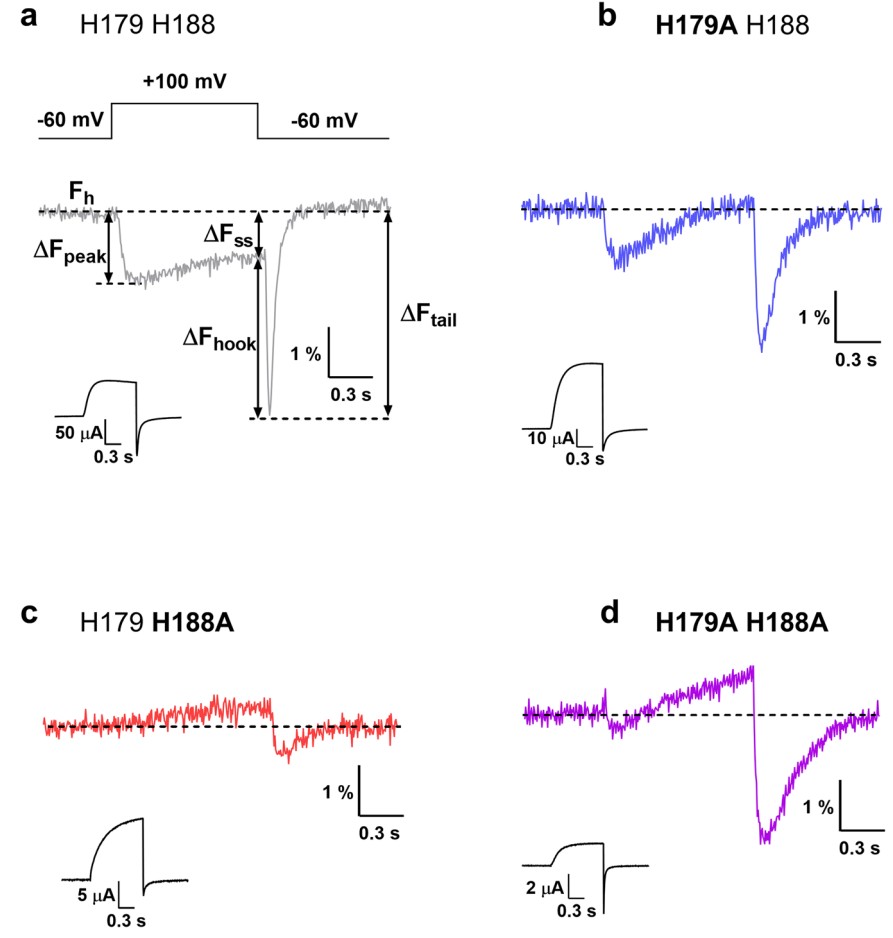

**Fig. 1 Fluorescence traces of CiHv1-E241C and alanine mutants.** Representative fluorescence responses and proton currents to a voltage step from −60 mV to +100 mV recorded from oocytes expressing Ci-Hv1 channels labeled with TAMRA-MTS at the E241C position (**a**) and different alanine mutants: H179A (**b**), H188A (**c**) and H179A H188A (**d**). $F_h$ is the fluorescence at the holding voltage of −60 mV, $\Delta F_{peak}$ is negative peak fluorescence change during the voltage step, $\Delta F_{ss}$ (steady state) is change in fluorescence at the end of the voltage step, $\Delta F_{tail}$ is maximum change in fluorescence during repolarization after the voltage step and $\Delta F_{hook} = \Delta F_{tail} - \Delta F_{ss}$. The insets show the measured proton current for the corresponding construct. Representative fluorescence traces were selected from N measurements: (**a**) $N = 72$, (**b**) $N = 13$, (**c**) $N = 8$, (**d**) $N = 28$.

may be close enough to the TAMRA-MTS attached to Cys at position 241. We identified two potential residues, both being histidines: H179 and H188 in the extracellular S1-S2 loop and at the top of S2, respectively (Fig. 2a, b), which could be sufficiently close to the moving TAMRA during the gating motions of S4. We mutated these two His residues to Ala (one by one and both at the same time) in the E241C background and examined the effect of these Ala mutations on the VCF signal (Fig. 1b–d). In addition, we examined whether Ala mutations have any effect on the conductance-voltage function of the ion channel, i.e. on the G-V function (Supplementary Fig. 2). Comparing the G-V functions measured for Ala mutations with the control G-V function (when there is no Ala mutation), no significant differences can be observed. (Supplementary Fig. 3j). In all Ala mutants, even in the double Ala mutant, the VCF signal remained detectable and retained its biphasic shape (Fig. 1d), arguing against the exclusive determinant role of His residues in signal generation. We also observed that at the end of the depolarizing pulse, the fluorescence intensity ($F_{ss}$) was higher/more positive in all alanine mutants compared to the CiHv1-E241C (Fig. 1), indicating that quenching by His at these positions does have a role in shaping the signal. The magnitude of the VCF signals did not change significantly as a result of the mutations, however, as it is influenced by factors such as the expression level of the construct or background fluorescence, direct comparison of magnitudes is

problematic. Therefore, we focused only on the shape of the VCF signals.

**Effects of Trp mutations.** Tryptophan is a more efficient fluorescence quencher than histidine for TAMRA and Alexa488 (Supplementary Fig. 1b and[4,39]). We therefore mutated the residues at the two critical positions, 179 and 188 to tryptophans, to test if the VCF signal is altered by the substitution. As with Ala, the Trp mutations did not cause significant changes in the G-V functions of the currents suggesting that the basic gating properties were preserved. (Supplementary Fig. 3). As expected, in both single mutants and the double mutant (Fig. 3) the shape of the signals was altered: the complex biphasic VCF signal shape disappeared and became monotonic. For both the Ala and Trp mutants, residue 188 seemed to be more influential in shaping the signal than residue 179, which is reasonable considering its shorter distance to the labeled 241C residue (Fig. 4). Although signal amplitudes cannot be compared directly due to possible differences in expression and background fluorescence levels, Trp mutant channels consistently produced signals of greater magnitude than variants carrying His or Ala residues. These results indicate that the presence of Trp residues at these positions contributes to the VCF signal generation process, but their role is not exclusive, since the VCF signal was still measurable even in

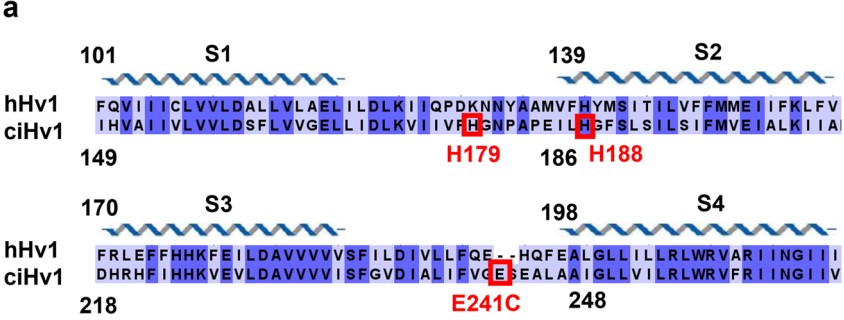

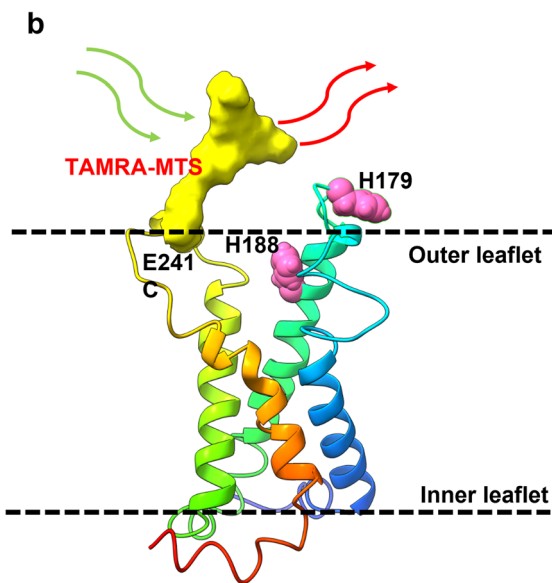

**Fig. 2 Sequence alignment and homology model of Ci-Hv1. a** Sequence alignment of S1–S4 voltage-sensing domains for Ci-Hv1 and hHv1. Conserved residues are highlighted with dark blue. Residue numbers are marked above for reference and the helical schematic above the alignment marks the extent of the helices in the homology model of Ci-Hv1, while residues that were mutated to Cys or Ala/Trp are indicated by red rectangles. Sequences included in the alignment are: Ciona intestinalis Hv1 (UniProtKB - Q1JV40), Human Hv1 (UniProtKB - Q96D96), visualized by Jalview[42]. **b** Homology model of the S1-S4 voltage-sensing domains for Ci-Hv1 based on mHv1 structure (PDB ID: 3WKV). Predicted domains are color coded as follows: S1 (blue), S2 (cyan), S3 (light green), S4 (orange). Membrane boundaries are estimated from superposition with the Kv1.2/2.1 paddle chimera structure where detergents and PCs (phosphatidylcholine) are resolved. TAMRA -MTS attached to E241C is represented in yellow, while the two histidine residues at position 179 and 188 are highlighted by magenta. This panel was created with UCSF Chimera[43] and the final image rendered with ChimeraX[44].

the double Ala mutant, in the absence of the quencher His or Trp residues. These observations imply the contribution of another mechanism, so we next investigated the possible role of the dye being transferred from a lipid phase to a more exposed aqueous phase during voltage-dependent conformational changes.

**Lipid effects on the VCF signal.** The fluorescence of many fluorophores (including Tetramethylrhodamine Maleimide, TMRM) is known to increase in a lipid environment[40]. Thus, we examined the effect of lipid (L-α-Phosphatidylcholine (PC)) addition on TAMRA-MTS (which has similar fluorescence properties as TMRM) in solution using a spectrofluorometer and we found an increase in fluorescence (Fig. 5a, b), similar to previously published TMRM data. A measurable effect was seen only in millimolar lipid concentrations, the emission intensity increased in a concentration dependent manner without a spectral shift (Fig. 5a, b) and significant increase was measured only at 5 mM PC.

The movement of the dye relative to the lipid membrane is expected to affect its fluorescence intensity and thus produce a VCF signal. We examined whether the presence of the lipid vesicles in the extracellular solution could affect the amplitude and/or the shape of the VCF signal. We assumed that if lipid vesicles were present in the extracellular solution during gating transitions, the TAMRA dye would detect a smaller change in the hydrophobicity of the microenvironment and as a consequence produce a smaller VCF signal. The vesicle size distribution of PC in the solution was measured with Dynamic Light Scattering at different PC concentrations. At 1 mM PC, the average size was $156.8 \pm 1.0$ nm, while at 5 mM PC it was $326.5 \pm 3.6$ nm (Supplementary Fig. 4).

Furthermore, we determined the intensity loss of both excitation and emission light due to scattering and absorption by the lipid (Supplementary Fig. 5). In our measurements, we observed a significant decrease in baseline fluorescence (Supplementary Fig. 5a). This is also reflected in the signal-to-noise ratio of the VCF signal even though the magnitude and shape of the

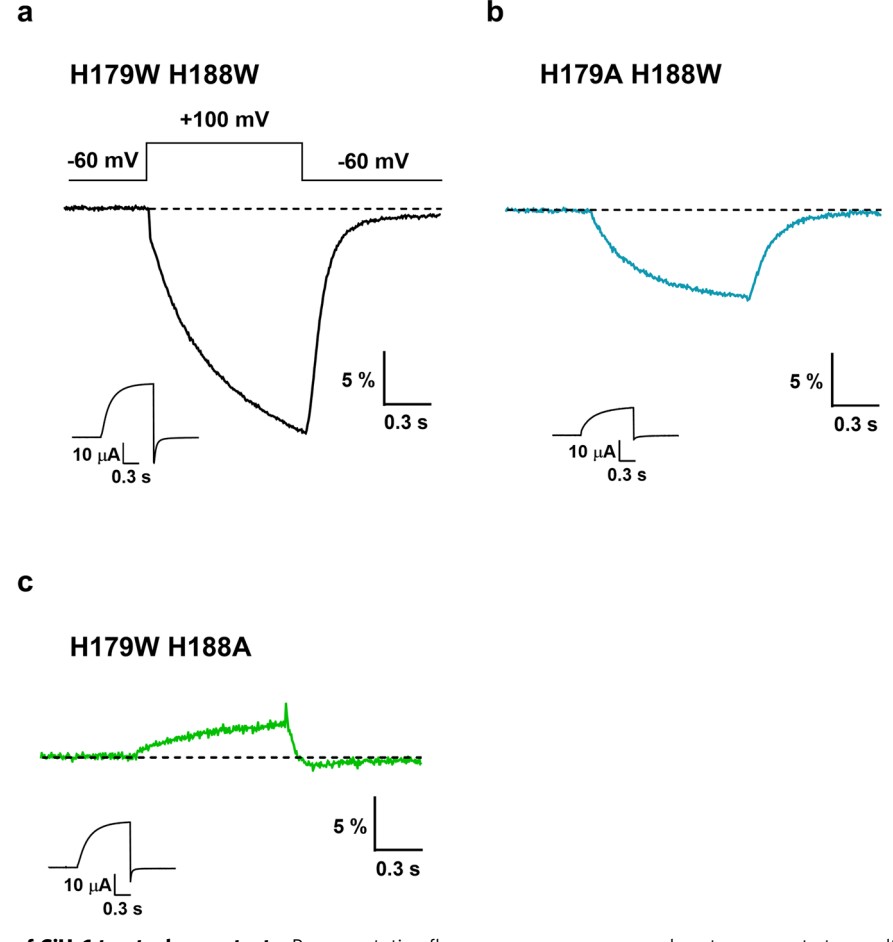

**Fig. 3 Fluorescence traces of CiHv1 tryptophan mutants.** Representative fluorescence responses and proton currents to a voltage step from −60 mV to +100 mV from oocytes labeled with TAMRA-MTS, expressing Ci-Hv1 E241C containing the following mutations: H179W/H188W (**a**), H179A/H188W (**b**) and H179W/H188A (**c**). The insets show the measured proton current for the corresponding construct. Representative fluorescence traces were selected from N measurements: (**a**) $N = 21$, (**b**) $N = 7$, (**c**) $N = 10$.

VCF signal is not changed by the lipid vesicles. Although light scatter and absorption by the lipid cause fluorescence intensity reduction, they do not change the shape of the VCF signal.

We first tested the lipid effect on the double Ala mutant lacking the quencher amino acids, as it was the simplest available system for this purpose. When the oocyte was in an extracellular solution containing 5 mM PC, the VCF signal amplitude of the double Ala mutant evoked by the voltage protocol was significantly reduced ($\Delta F_{tail}/F_h$, normalized to PC-free value: $0.41 \pm 0.13$, $p = 0.02$, Fig. 5c–e). The effect was reversible as indicated by panel (d) and the last column of panel (e) (wash, $0 \mu M$). Smaller lipid concentrations caused no significant change in the signal amplitude (Fig. 5e).

Next, we performed the same experiment on the double Trp mutant. In this case, the addition of lipid vesicles would be expected to have a smaller effect because of the strong contribution of the quenching Trp residues to the generation of the VCF signal. In other words, the dominance of Trp quenching should mask the effect of PC in the environment of the dye resulting in a smaller change of the VCF signal than in the double Ala mutant. As expected, the addition of the PC did not cause any significant change either in the amplitude or the shape in the VCF signal, supporting our assumption ($\Delta F_{tail}/F_h$, normalized to PC-free value: $1.04 \pm 0.10$, $p = 0.42$, Fig. 5f–h).

Third, we tested the effect of lipid vesicles on CiHv1-E241C. Two quenching His residues are present in this channel partially

contributing to the formation of the VCF signal, so the transitioning of TAMRA between environments of different PC concentrations should not exclusively determine the signal generation in the CiHv1-E241C construct. Therefore, the effect of lipid vesicles is expected to be smaller than for the double Ala mutant but greater than for the double Trp mutant. Addition of 5 mM lipid vesicles confirmed our expectation as the signal amplitude was not significantly reduced ($dF_{tail}/F_h$, normalized to PC-free value: $0.76 \pm 0.07$, $p = 0.33$) (Fig. 5i–k).

Based on these results, we conclude that a change in the hydrophobicity of the fluorophore environment may be sufficient to generate the VCF signal observed for the double alanine mutant lacking quenching residues (lipid signal). In contrast, in the CiHv1-E241C and double tryptophan constructs, quenching amino acids (His and Trp) along with the change in environmental hydrophobicity together shape the VCF signal (amino acid + lipid signal).

We also investigated whether extracellularly applied lipid molecules affect the proton current and found that there was no significant effect on the current of any of the CiHv1 constructs (Supplementary Fig. 6).

**Fluorescence traces of heterodimer CiHv1 constructs.** Hv1 is known to function physiologically as a dimer, thus it is conceivable that the VCF dye attached to one subunit of the dimer is

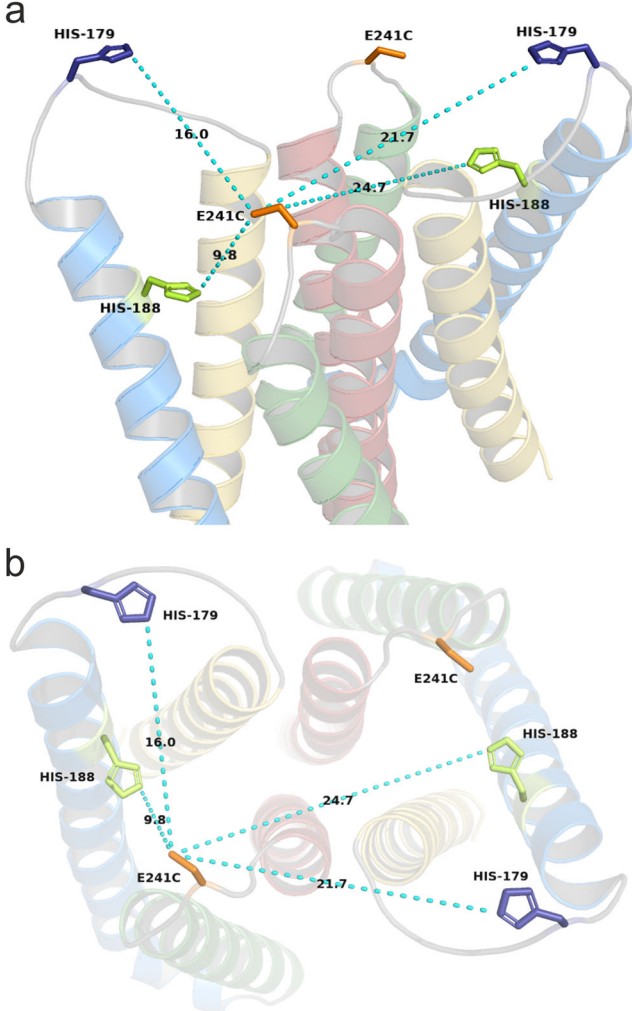

**Fig. 4 Distances in the homology model of Ci-Hv1.** The homology model of the S1-S4 voltage-sensing domains for Ci-Hv1 is based on the reported model[45]. Helices are color coded as follows: S1 (yellow), S2 (blue), S3 (green), S4 (red). E241C mutation is highlighted in orange, histidine residues at position 179 is represented in blue, while at position 188 is light green. **a** panel shows a side-view, (**b**) panel is a top-view. The light blue dashed lines represent the distances between E241C and His residues in the same subunit and in the other subunit. Black numbers on the light blue dashed lines show the distance in angstroms. These panels were created with Pymol[46].

quenched by a residue situated on the other subunit. Such interaction was shown previously for tandem dimers of Hv1[3]. Therefore, we performed experiments to determine whether the quenching effect is dominantly due to residue(s) from the same subunit of the CiHv1 dimer to which the TAMRA dye is linked, or whether quenching amino acids from the other subunits also have a significant effect on the VCF signal.

We have mixed RNA of E241C subunits (no Trp) with H179W/H188W subunits (no Cys = no dye) in a 20–80% ratio, respectively. In this way, in the majority (89%, based on the binominal distribution) of dimer Hv1, which can produce a VCF signal, the TAMRA-labeled subunit was paired with a subunit which does not contain labelable Cys, but has two Trp residues. The resulting VCF signal (Fig. 6a–c) is very similar to the one measured for CiHv1-E241C (Figs. 1a and 5i). This leads to the

conclusion that the dominant quenching residue must be located on the same subunit as the TAMRA dye.

We have repeated the same RNA-mixing experiment with the double Ala construct: 20% RNA of E241C subunits (no Trp) and 80% RNA of H179A/H188A subunits (no Cys = no dye). The measured VCF signals (Fig. 6d–f) were almost identical to the double Trp mixed-RNA ones, further supporting our conclusion that the dominant quenching amino acids are located on the same subunit as the TAMRA dye. When lipid was added to the external solution during the mixed-RNA measurement of H179A/H188A, no significant change was observed in the magnitude of the VCF signal. This is in good agreement with results obtained for homomeric CiHv1-E241C with $2 \times 2$ His residues, since in this case the majority of the VCF signal originates from TAMRA molecules in the vicinity of His residues on the same subunit, being farther from the Ala residues on the other subunit. These observations support the idea that residues on the neighboring subunit do not influence the lipid effect.

**Exclusion of the electrochromic effect.** The VCF signals of all constructs had a rapid component upon depolarization, but it was especially prominent in the Trp mutants. We examined whether the rapid component could be created by the electrochromic effect, or in other words the Stark shift. The properties of electrochromic effect are a nearly linear $\Delta F/F$ - V relationship and rapid kinetics matching the kinetics of voltage clamping[7–9]. Thus, we compared the kinetics of the signal with the clamping speed of our recording setup. During our VCF recordings, applying voltage step protocols, the voltage change develops with a sub millisecond time constant ($0.33 \pm 0.08$ ms, from $-60$ mV to $+100$ mV), while the rapid component of the double Trp construct signal is in the millisecond range ($1.03 \pm 0.24$ msec, from $-60$ mV to $+100$ mV), which is significantly greater ($p = 0.026$, $N = 4$). Regarding the VCF signal of CiHv1-E241C, the rapid component is in the 10-millisecond range ($14.6 \pm 4.2$ ms, from $-60$ mV to $+100$ mV, $N = 4$), which is also significantly greater ($p = 0.014$). Based on these comparisons we exclude the possibility of the Stark-shift being responsible for the fast fluorescence component.

**Discussion**
The goal of this study was to explore the mechanisms contributing to the generation of the VCF signal of Ci-Hv1 and to draw conclusions about its conformational changes. In addition, we aimed at identifying mechanisms that may be generally true for VCF signal formation, thus facilitating the interpretation of VCF signals measured on other voltage-gated proteins.

We found earlier that a His residue in the extracellular part of TMEM266 plays a role in the formation of the VCF signal[4]. This is consistent with the assumption that quenching by certain amino acids may contribute to VCF signal generation. In the extracellular part of Ci-Hv1, we found two potential quenching residues: H179 and H188 (Figs. 2 and 4). The mutation of these two histidines to non-quenching residues (Ala) (Fig. 1) and to strong quenchers (Trp) (Fig. 3) indicated that amino acids at these positions play a role in the VCF signal formation as quenchers of TAMRA. However, since the VCF signal remained detectable even when both histidines were replaced by alanines (Fig. 1a), we hypothesized that there must be another contributing mechanism in the background: the transitioning of TAMRA between lipid and aqueous environments. Indeed, the fluorescence intensity of free TAMRA increases when lipid vesicles are added to the solution (Fig. 5a, b), suggesting that TAMRA preferentially partitions into amphipathic environments where it

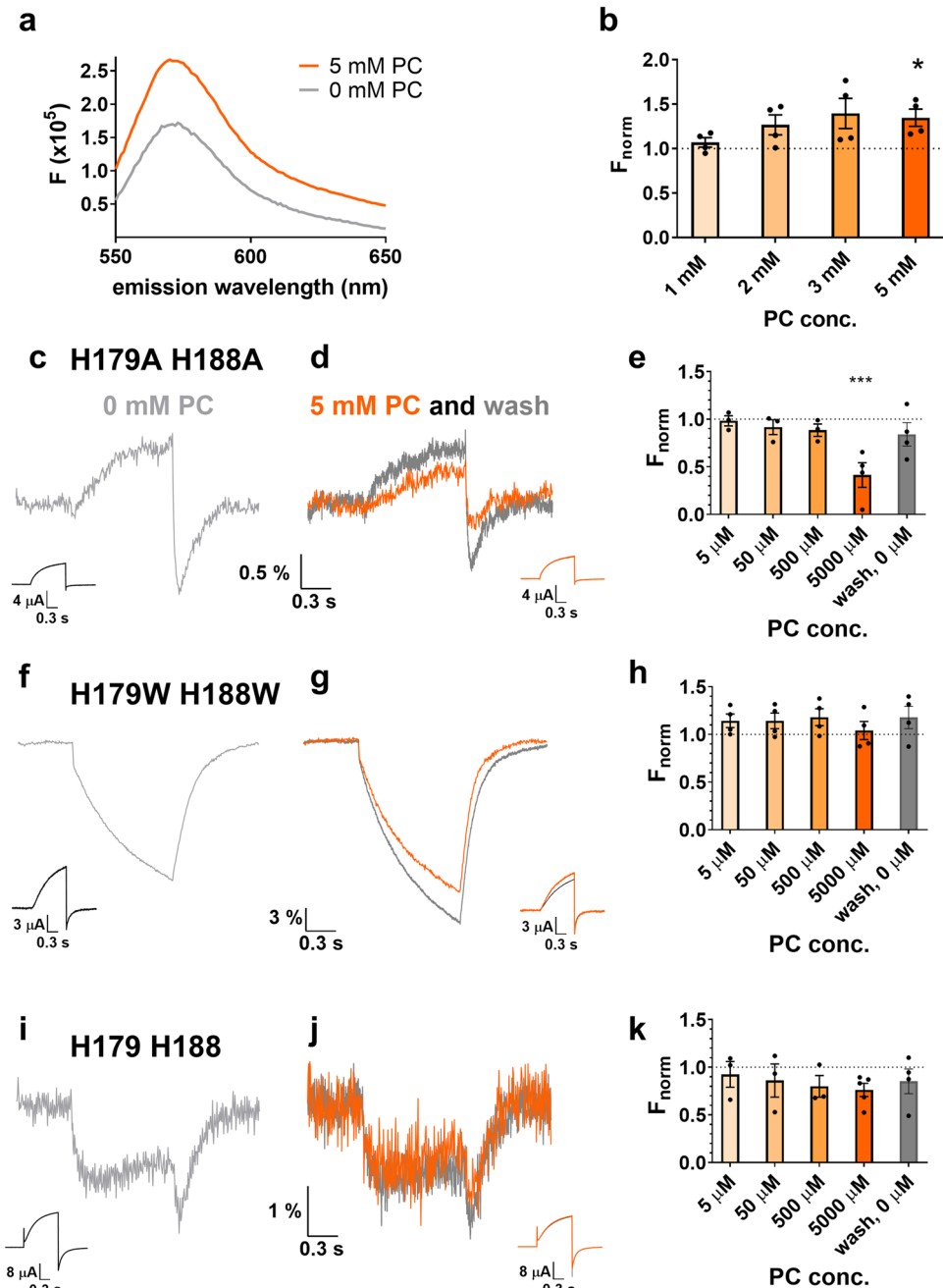

**Fig. 5 Effect of PC (phosphatidylcholine) on TAMRA-MTS fluorescence. a** Effect of 5 mM PC (phosphatidylcholine) on the emission spectrum of TAMRA-MTS in an aqueous solution, in a cuvette (excitation wavelength: 535 nm). **b** TAMRA-MTS fluorescence in solution in the presence of PC at different concentrations normalized to the control solution (0 mM PC) measured at the wavelength of maximum emission ($N = 5$). Representative fluorescence responses and proton currents to a voltage step from $-60$ mV to $+100$ mV from oocytes labeled with TAMRA-MTS on E241C, expressing the constructs Ci-Hv1 H179A/H188A (**c, d, e**), H179W/H188W (**f, g, h**) and H179/H188 (**i, j, k**). **c, f, i** panels show the control VCF signals in the absence of PC, before PC application, while the (**d, g, j**) panels show VCF signal in the presence of 5 mM PC (orange) and after wash (dark grey). Bar charts in panels **e, h, k** summarize the average PC effects at different concentrations along with washout. Vertical axis shows $\Delta F_{tail}/F_h$ values, normalized to the control value (0 mM PC). Error bars represent SEM, *indicates significant difference (*$p < 0.05$, ***$p < 0.001$) compared to the control value using one-way ANOVA. ($N = 3$ at least for **e** and **k**; $N = 4$ for **h**).

is more fluorescent[40]. Thus, conformational changes modifying the environmental polarity of the region surrounding the S3–S4 loop could readily produce fluorescence changes from TAMRA covalently attached to it. In the absence of a strong quenching residue, the VCF signal is markedly reduced when lipid vesicles are added to the extracellular solution. This would not be expected if TAMRA remained buried in the cell membrane and suggests instead that the linker attaching TAMRA to the S3-S4

loop allows it to project far enough out from the cell membrane to directly interact with lipid vesicles in solution.

For the majority of constructs the VCF signal is clearly biphasic with a fast, initial decrease in fluorescence followed by a second, slower component. Thus, the channels must undergo at least two transitions between a minimum of three distinct conformations. To see if this minimal description is sufficient, we fitted the measured fluorescence to a sequential 3-state

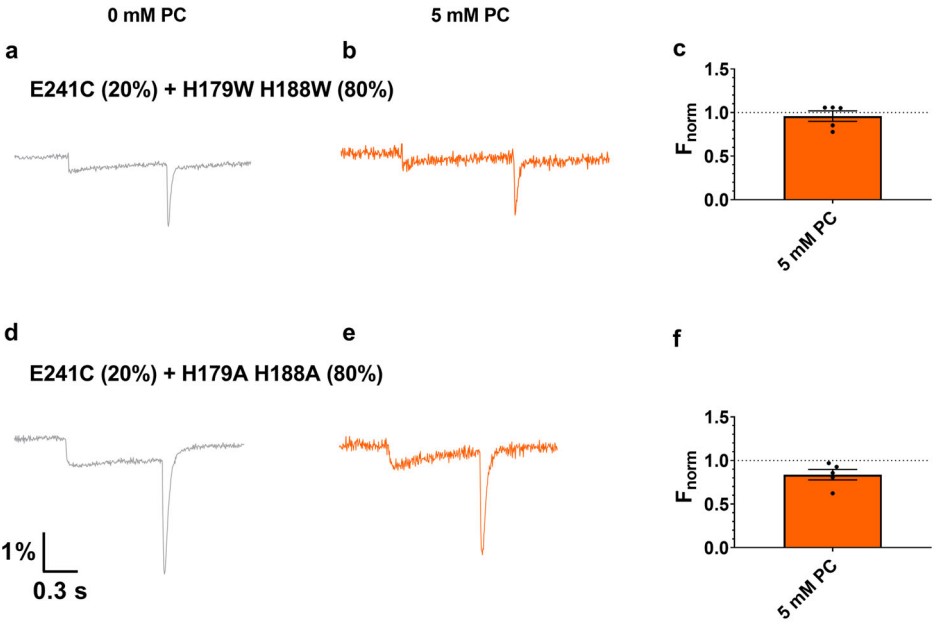

**Fig. 6 Fluorescence traces of heterodimer CiHv1.** Representative fluorescence responses to a voltage step from −60 mV to +100 mV from oocytes labeled with TAMRA-MTS on E241C in only one subunit of the dimer. The TAMRA-labeled subunit was paired with a subunit which does not contain labelable Cys but two Trp residues (**a**, **b**, **c**) or two Ala residues (**d**, **e**, **f**) by mixing RNA of E241C subunits (no Trp) with H179W/H188W or H179A/H188A subunits (no Cys) in a 20–80% ratio, respectively. **a**, **d** panels show the control VCF signals in the absence of PC, while the (**b**, **e**) panels in the presence of 5 mM PC in the extracellular solution from the same oocyte. Bar charts in panel **c**, **f** summarize the average PC effects at 5 mM concentration. Vertical axis shows $\Delta F_{tail}/F_h$ values, normalized to the control value (0 mM PC). Error bars represent SEM ($N \geq 4$).

model (Resting (1) ↔ Intermediate (2) ↔ Activated (3)) similar to the one suggested by the Larsson group[35] (Figs. 7 and 8). For a given construct/condition, the fluorescence intensity of individual dye molecules is assumed to depend only on the protein state (1, 2, or 3), and so at any point in time the observed fluorescence intensity depends upon the fraction of channels in each state (p1, p2, and p3) and the fluorescence intensity when a channel is in that state (F1, F2, F3). Initially at the resting potential, most channels are in state #1 (Fig. 7 left panels). Upon depolarization channels undergo the transition #1 → #2, and the fluorescence of the individual dye molecules decreases (F2 < F1). The lower fluorescence of state #2 is consistent with exposure of TAMRA to a more polar environment in this intermediate state as it moves away from the membrane (Fig. 7 middle panels). During this transition, the contribution of quenching residues to the signal is negligible. For strong depolarizations, channels progress from state #2 to state #3 and the fluorescence approaches a steady-state value (Fig. 7 right panels). Upon repolarization, the channels transition from #3 → #2 and the fluorescence decreases quickly, then channels return from #2 → #1 and the fluorescence increases slowly. As shown in Fig. 8, this simple 3-state model fits the VCF signal of all constructs well. In all constructs when quencher amino acids are present, the VCF signal during the transition to state #3 is shaped by the combined effect of the dye approaching the membrane and at the same time the quenching residues. For the CiHv1-E241C, H179A, H188A (weak quenchers) and H179A/H188A (no quencher) the lipid effect dominates, and the microscopic fluorescence (for a single TAMRA molecule) as well as the macroscopic fluorescence (for the whole oocyte) increase close to the initial fluorescence value, (see Supplementary Table 1, Fluorescence values for state 1 and 3, except H179A H188W). Conversely, for the constructs containing Trp at critical positions the macroscopic fluorescence continues to

decrease along with the microscopic fluorescence (F3 < F2) (Supplementary Table 1, Fluorescence values of H179A H188W for state 1 and 3) indicating that the strong quenching by Trp overrides the lipid effect.

The more dominant quenching residue at position 188 should be on the same subunit of the dimer CiHv1 as the TAMRA dye because of the following three results: (I) the shortest distance based on the homology model of Ci-Hv1 is between E241 and H188 in the same subunit (Fig. 4.). (II) The similarity of homodimer (Figs. 1a and 5i) and heterodimer CiHv1 E241C VCF signals (Fig. 6). (III) The non-significant lipid effect on heterodimer CiHv1 E241C VCF signals where the TAMRA-labeled subunit was paired with a subunit, which does not contain labelable Cys but two Ala residues (Fig. 6d–f). The observed effect was similar to the one seen on homomeric CiHv1 E241C channels containing only His residues.

The minimal 3-state description of the VCF signals is clearly sufficient although multi-step fluorescence kinetics could potentially also arise from a 2-state model where other quenching mechanisms (e.g. a non-voltage dependent transition of the protein and fluorophore) could contribute to the signal. However, because the steady-state fluorescence has a non-monotonic dependence on voltage, this system cannot be described by a model with a single voltage-dependent transition. Instead, to describe the kinetics and voltage-dependence of the fluorescence signals any model must have at least two voltage-dependent transitions, and the 3-state model is the simplest possible example of such a model. The Larsson group used the same fluorescence intensity relationships (regarding F1, F2, F3)[35], but incorporated the dimer structure of Hv1 into their model to include three transitions. These were the independent activation of the two VSDs and the subsequent cooperative opening transition, and thus effectively four states. Based on the analysis of Hv1 gating currents Gonzalez et al.

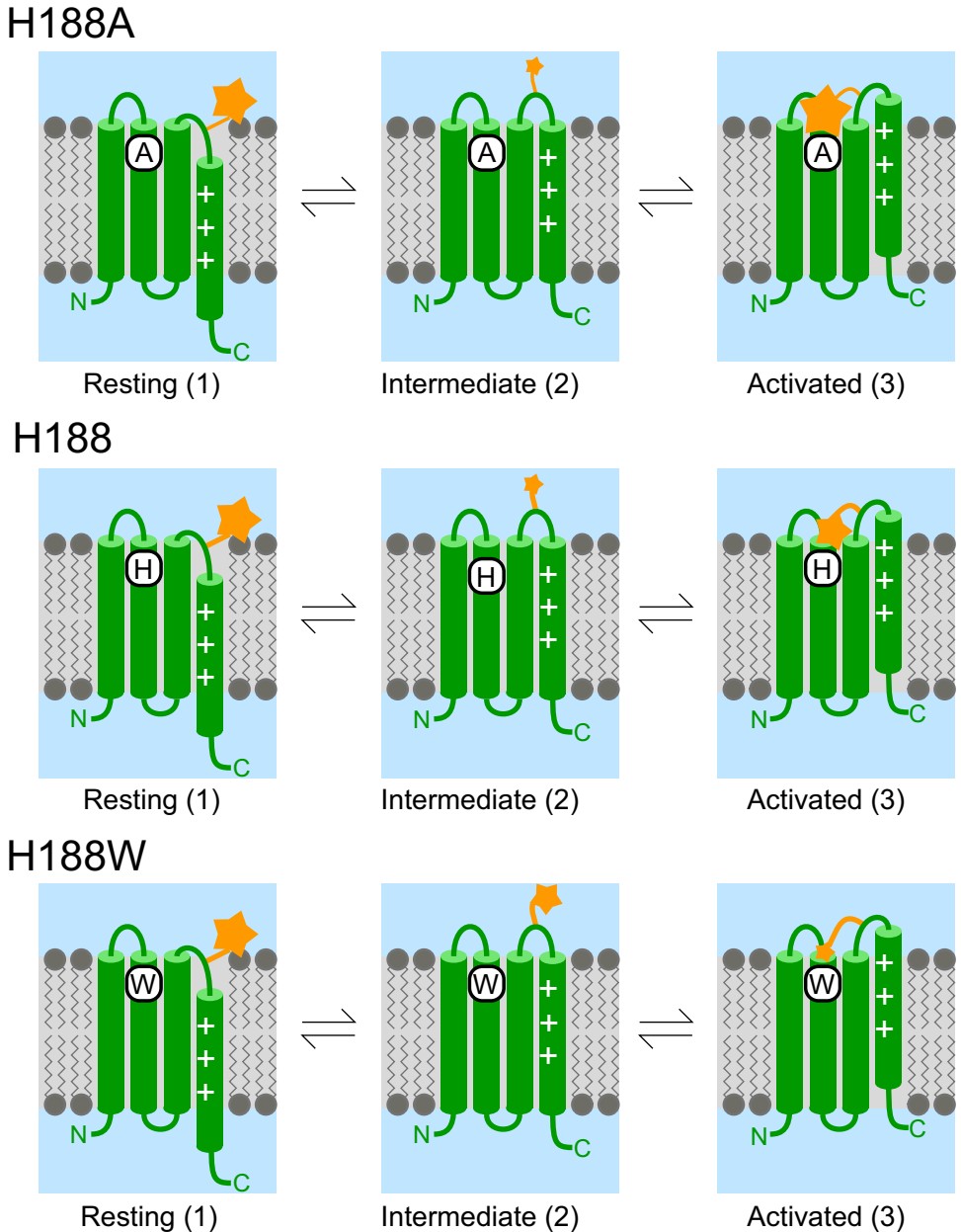

**Fig. 7 Model of VCF signal generation during CiHv1 conformational changes.** The model assumes two major VSD movements among three states traced by the attached TAMRA (orange star). The resting (1), the intermediate (2) and the activated (3) states are associated with different fluorescence intensities (F1, F2, F3) in each construct for a single TAMRA molecule. In the H188A/H179A mutant (top row panels) there is no quenching residue near TAMRA. In response to a depolarizing pulse, the dye first moves away from the lipid bilayer from state 1 to 2 and the fluorescence intensity decreases, since F1 > F2. Subsequently, there is a second motion from state 2 to 3, when the dye approaches the lipid bilayer, and the fluorescence increases to a level above the initial one: F3 > F1 > F2. In single alanine mutants and the CiHv1-E241C there is/are weak quenching residue(s) near the dye (middle row panels). The motion is similar to that in the top row panels, but during the second motion as TAMRA approaches both the quencher and the bilayer, the latter effect overrides the weak effect of the quencher, thus, the fluorescence increases but remains below the original one: F1 > F3 > F2. In the tryptophan mutants (bottom row panels) there is/are strong quenching residue(s) near the dye. During depolarization, it moves similarly to the top row panel motion, away from the membrane and toward the quenching residue from state 1 to 2 and the fluorescence intensity decreases, F1 > F2. Subsequently, in the second motion from state 2 to 3, TAMRA approaches both the membrane and the quencher, the latter effect overrides the lipid effect and fluorescence further decreases: F1 > F2 > F3.

proposed a more refined 5-state model for gating[37]. However, it should be noted that gating current measurements provide different types of information about conformational changes in VSD than VCF. Whereas gating currents detect any sufficiently rapid movement of charge (e.g., displacement of S4 with respect to the charge transfer center), a fluorescence reporter will only detect conformational changes that impact the surroundings of

the fluorophore. However, while fluorophores may miss or blend some transitions, they can detect slow (and even voltage-independent) changes and can also provide extra information about the 3-D movement of the VSD. Although mechanistically our model is not as detailed as the presently accepted best gating models of Hv1, it is consistent with them and explains VCF signal generation considering both parallel and

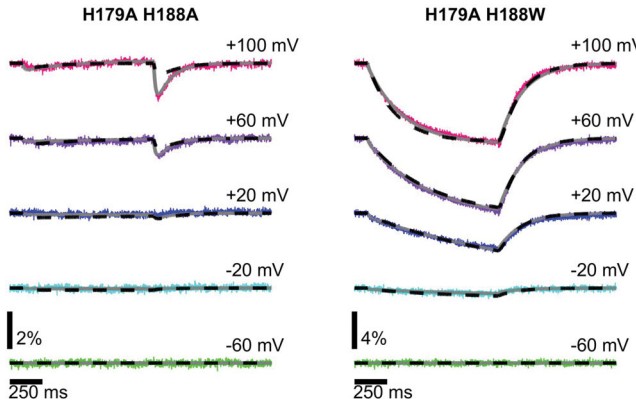

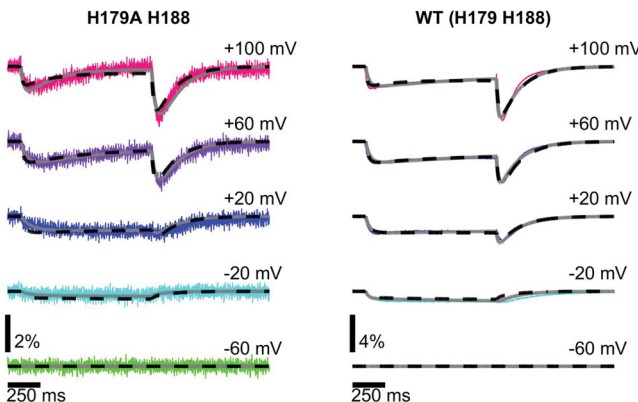

**Fig. 8 Experimental and model traces of CiHv1 VCF signals.** Fluorescence responses to voltage steps from −60 mV to +100 mV (pink), +60 mV (purple), +20 mV (dark blue), −20 mV (light blue) and −60 mV (green) recorded from oocytes expressing CiHv1 channels labeled with TAMRA-MTS at the E241C position in WT (H179/H188) and different mutants: H179A/H188A, H179A/H188Wand H179A/H188. The fluorescence signal from these four constructs can be simultaneously described by a single, common three-state kinetic model (dashed black lines), although allowing for kinetic differences between the constructs (grey lines) further improves the agreement between the experiment and model.

perpendicular movement of the VSD to the plane of the membrane.

Quenching residues have been previously identified as contributors to the VCF signal in voltage-sensitive proteins. For example, TMEM266 has a histidine at the top of S2 (H139), which corresponds to H188 in Ci-Hv1. A histidine is also present in human Hv1 at this position, but the other histidine (corresponding to H179) is missing. This may explain why it is more difficult to measure a VCF signal for human Hv1 compared to Ci-Hv1: only one such publication is known[36]. There is, however, another His residue near the top of S4, H193, which is only two positions away from the inserted cysteine that was labeled for VCF measurements (F195C). Being so close to the dye, the distance to H193 would unlikely change by a great extent during gating, so it would not efficiently generate a signal. Based on this, the VCF signal in human Hv1 may consist almost only of the lipid signal. In the Shaker K⁺ channel, there are two quenching residues in the extracellular portion of the VSD (H254 and Y255), and six others in the S5-S6 loop (Y442, W461, W462, M467, Y472, M475), which explains the large variety of VCF signal shapes and positions at which the attached dyes yield a signal[41].

However, based on our results, extra content can be retrospectively associated with VCF signals obtained from these proteins if dye movement relative to the lipid membrane is also taken into consideration.

## Materials and methods

**Molecular biology, expression systems.** The plasmid encoding the *Ciona intestinalis* voltage-gated proton channel (Ci-Hv1) containing the cysteine mutation E241C was generously provided by Peter Larsson (Univ. Miami). Other mutations (H179A, H179W, H188A, H188W) were constructed via site-directed mutagenesis using an overlapping PCR procedure and the mutations were verified by sequencing. DNA was linearized with *SacI* and transcribed to RNA with the Invitrogen Transcription Kit (ThermoFisher, Waltham, MA). *Xenopus laevis* oocytes for voltage-clamp fluorometry experiments were purchased from EcoCyte Bioscience (Dortmund, Germany). Oocytes were injected with 50 nl of RNA at a concentration of 1 μg/μl and incubated at 18 °C for 2–5 days in ND93 containing 93 mM NaCl, 5 mM KCl, 1.8 mM CaCl₂, 1 mM MgCl₂, 5 mM HEPES and 50 mg/l Gentamycin, pH 7.4. Chemicals used for the preparation of the solutions were purchased from Sigma–Aldrich (St. Louis, MO).

**Voltage-clamp fluorometry (VCF).** For VCF recordings oocytes were labeled for 20–40 min, at 13 °C with 10 μM of 2-((5(6)-tetramethylrhodamine)carboxylamino) ethyl methanethiosulfonate (TAMRA-MTS, Toronto Research Chemicals, Toronto, ON, Canada), diluted in ND93 solution. 50 nl of 1 M HEPES (pH = 7.0) was injected into each oocyte prior to labelling to minimize pH changes due to the proton efflux during recording. The extracellular solution for Ci-Hv1 recordings contained (in mM) 75 NaCl, 2 CaCl₂, 1 EGTA and 100 HEPES, pH 7.5 and the intracellular solution contained 3 M KCl. After labeling, oocytes were extensively washed in ND93 and stored in the dark, at 13 °C prior to performing experiments. Ionic currents were recorded with an Oocyte Clamp OC-725C amplifier (Warner Instruments, Hamden, CT). VCF signals were acquired through a 40×, 0.8-NA CFI Plan Fluor Nikon fluorescence water-immersion objective on a Nikon Eclipse FNI microscope (Nikon, Tokyo, Japan) and a photodiode (PIN-040A; United Detector Technology, OSI Optoelectronics, Hawthorne, CA). Illumination was provided by a green LED (530 nm), M530L2-C1 from ThorLabs (Newton, NJ). TAMRA-MTS signals were recorded using a 545/25 excitation filter, a 565LP dichroic mirror and a 605/70 emission filter. The signal from the photodiode was recorded by an Axopatch 200 A amplifier and a Digidata-1550 digitizer controlled by pClamp10 (Molecular Devices, San Jose, CA). Hv1 channels were activated by 1-s-long depolarizations to +100 mV from a holding potential of −60 mV to evoke the VCF signals. For lipid measurements, L-α-Phosphatidylcholine (PC) (P5638 Sigma–Aldrich) was dissolved in the recording solution at 100 mM concentration (stock solution), vortexed vigorously, and then diluted to the appropriate concentration (5000-, 500-, 50-, 5 μM). There was no incubation time, the effect of lipid was measured immediately after the solution change in the recording chamber.

**Spectrofluorometry.** TAMRA-MTS was excited at 535 nm and the emission spectrum was measured in the range of 550–650 nm, in the absence or presence of PC using a Spex Fluoromax (Jobin Yvon) spectrofluorometer. The recording solution was the same as the extracellular solution for Ci-Hv1 recordings, contained (mM) 75 NaCl, 2 CaCl₂, 1 EGTA and 100 HEPES, pH 7.4, and the PC was dissolved in this solution. TAMRA-MTS concentration was 20 nM. Measurements were performed at room temperature with continuous stirring, right after mixing PC with TAMRA-MTS (within 1 min). For control recordings, we measured the emission spectrum of the extracellular solution without TAMRA-MTS but with PC at different concentrations. These control recordings were subtracted from the appropriate measurement data and plotted in the figures.

**Spectrophotometry.** Light absorbance of PC was measured with a NanoDrop 1000 Spectrophotometer at different PC concentrations, in the wavelength range of 260 nm and 700 nm. The recording solution was the same as the extracellular solution for Ci-Hv1 recordings, contained (mM) 75 NaCl, 2 CaCl₂, 1 EGTA and 100 HEPES, pH 7.4, and the PC was dissolved in this solution. Measurements were performed at room temperature. For control recordings, we measured the absorbance of the PC free extracellular solution. $A = -\log(I / I_{ctrl})$, where A is the absorbance, $I_{ctrl}$ is the control light intensity measured without PC and I is the light intensity measured at different PC concentrations. The "decrease in fluorescence" at different PC concentrations was calculated as $1 - 10^{-A_{540}} \times 10^{-A_{565}}$, where $A_{540}$ and $A_{565}$ are the absorbance at 540 nm and 565 nm, at the maximum of TAMRA-MTS excitation and emission spectra, respectively.

**Dynamic light scattering.** The hydrodynamic size and size distribution of the lipid vesicles were determined by dynamic light scattering (DLS). The samples were filtered through a 0.45 μm syringe filter to eliminate dust or other impurities. DLS measurements were performed using a Zetasizer Nano ZS (Malvern Instruments, Malvern, UK) equipped with a He-Ne laser (633 nm) at 25 °C and at a detector position of 175°. The Z-average size and polydispersity index (PDI) were calculated

by cumulants analysis. The distribution of particle sizes was determined by multiple exponential fit.

**Data analysis.** Electrophysiological data analysis was performed using Clampfit (v10; Molecular Devices), SigmaPlot (v10; SigmaStat Software, San Jose, CA) and Excel (Microsoft, Redmond, WA). Current and VCF signals were recorded at 2 kHz and low-pass filtered at 1 kHz. Fluorescence traces represent single recordings (without averaging) and were filtered with a Boxcar filter (smoothing points: 3). The magnitude of the VCF signals was expressed as $\Delta F/F_h$ in percentage, where $F_h$ is the baseline fluorescence at the holding voltage of $-60$ mV and $\Delta F$ is the change in fluorescence upon depolarization. Four different $\Delta F$ values can be distinguished (Fig. 1a): $\Delta F_{peak}$ is the difference between $F_h$ and the negative fluorescence peak during depolarization, $\Delta F_{ss}$ is change in fluorescence at the end of the voltage step relative to $F_h$, $\Delta F_{tail}$ is the difference between $F_h$ and the negative fluorescence peak during repolarization, and $\Delta F_{hook}$ equals ($\Delta F_{tail} - \Delta F_{ss}$).

To correct for photobleaching, the baseline fluorescence trace, which has no change in voltage, was fitted with an exponential function and the parameters (amplitude and tau) of this fitting was used for photobleaching correction. F-V plots show the $\Delta F/F_h$ values according to the different $\Delta F$ calculation methods, normalized to the maximum value as a function of test potential. G-V curves were constructed by using the leak-corrected tail current measured during repolarization, normalized to the maximum value, plotted as a function of test potential.

**Statistics and reproducibility.** Bar charts show mean values with individual data points. Reported errors are SEM, numbers of cells (N) involved in the given analysis are shown in the text or in the figure. P values were calculated based on ANOVA (one way) analysis with a Holm-Sidak post-hoc test. Differences were considered significant (asterisk, *) when $p < 0.05$. Measurements were taken from distinct samples (regarding Spectrofluorometry, Spectrophotometry and Dynamic light scattering) or from different oocytes (regarding VCF).

**Modeling.** Fluorescence intensity at a given time point was calculated as the following: $F = \sum_{i=1}^3 P_i F_i$, where $P_i$ is the probability being in state "i" and $F_i$ is the fluorescence intensity in state "i". The transitions between states "i" and "j" were assumed to pass via an intermediate transition state "ij" so the transition rate was then $r_{ij}(t) = \exp(-(\Delta g_{ij} + \Delta q_{ij} V(t))/kT)/\tau$, where $\Delta g_{ij} = g_{ij} - g_i$ is the free energy difference at $V = 0$ mV between states "i" and "ij", $\Delta q_{ij} = q_{ij} - q_i$ is the charge transferred across the membrane in moving from state "i" to "ij", k is Boltzmann's constant, T is the temperature, and $\tau$ characterizes the timescale for a barrier free reaction. The fluorescence intensities ($F_i$), free energy ($g_i$, $g_{ij}$), and charge ($q_i$, $q_{ij}$), were then estimated via a non-linear, least squares fit to the experimental fluorescence measurements. The simplest model that adequately fitted allowed sequential transitions between $N = 3$ states (i.e. $1 \leftrightarrow 2 \leftrightarrow 3$) (See the parameters of model calculations in Supplementary Table 1).

**Reporting summary.** Further information on research design is available in the Nature Research Reporting Summary linked to this article.

## Data availability
The datasets generated and analyzed during the current study are available from the corresponding author on reasonable request.

## Code availability
The computer code and algorithm used to generate results that are reported in the paper are available from Gilman E. S. Toombes on reasonable request.

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

## Acknowledgements

We thank Peter Larsson for the Ci-Hv1 cDNA and members of our Electrophysiology group and Ion channel structure-function laboratory for helpful discussions. This work was supported by National Research Development and Innovation Office: 2019-2.1.11-TÉT-2019-00006 (F.P.), 2019-2.1.11-TÉT-2019-00059 (Z.V.), OTKA 132906 (Z.V.); by OTKA Bridging Fund 1G3DBKB0BFPF247 (F.P.); by János Bolyai Research Scholarship of the Hungarian Academy of Sciences (BO/00355/21/8) (F.P.); by the ÚNKP-21-5-DE-460 (F.P.) and ÚNKP-22-3-II-DE-38 (A.F.) New National Excellence Program of the Ministry for Innovation and Technology.

## Author contributions

F.P., J.A., Á.K. and A.F. performed the experiments. F.P., V.Z., A.F., G.E.S.T. and Gy.P. wrote the manuscript. F.P., G.E.S.T., Z.P., A.F., J.B., J.A., Á.K. and S.K. analyzed the data and made figures. F.P. and A.F. performed the statistical analysis. A.B., Z.P. and F.P. did the molecular biology work. G.E.S.T. performed the model calculation. F.P. and Z.V. designed the experiments and revised the manuscript. All authors reviewed the manuscript.

## Funding

## Competing interests

The authors declare no competing interests.
