## [Peer Review File · Communications Biology]

Reviewers' comments:

Reviewer #1 (Remarks to the Author):

In this work, Papp and colleagues investigate the activation mechanism of the voltage-gated proton channel Hv1 using the voltage-clamp fluorometry (VCF) approach. This is an experimental technique that allows one to track the conformational changes of a protein, as an attached fluorescent probe is differentially quenched at various protein states. The optical signal is sometimes composite of several quenching processes, making it difficult to interpret. Papp and colleagues investigate several different mechanisms that underlie the fluorescence signal, to understand better how the Hv1 channel can activate.

Several novel ideas arise from this work. The authors show how histidine residues can be fluorescence quenchers during VCF experiments and provide an experimental means to test the effect of lipids on fluorescence signals. An important advantage of their work is that it can provide a straightforward structural interpretation of the VCF findings. As more protein atomistic structures are increasingly made available by the cryo-EM revolution, VCF is one of the few approaches that can add a dynamic perspective to these structures during conformational changes pertinent to their function; so improving our ability to interpret VCF signals in terms of protein structure is highly valuable.

The work is of high experimental quality, but the conclusions require some additional analysis and control experiments. I believe they should be feasible for the authors.

First, the fluorescence signals are generally shown as the amplitude of fluorescence change over background fluorescence (DF/F). Oocytes can have a large variability of expression level, which would in turn affect the amplitude of the signals. Likewise, background fluorescence can be affected by oocyte-to-oocyte variability of animal pole pigmentation (dark oocytes allow for less autofluorescence). A better way to quantify DF is normalization by maximal G , or (assuming the channel voltage dependence is not affected by the mutations) current at a saturating membrane potential. This will clarify whether DF are (for example) small because the fluorophore is quenched less, or there is less expression in that oocyte / channel clone, or the oocyte had higher autofluorescence. This is particularly important since DF amplitude is used in subtractions, and more advanced mathematical fitting.

The authors assume that the fluorescence signals arise from the interaction of the fluorophore with quenching groups in the same Hv1 subunit. But could they also arise from interactions with residues in a neighboring subunit, either nearby or associated as a dimer? For example, Mony et al (Nat Struct Mol Biol, 2015) showed that a fluorophore in S1 can be quenched by a Trp in S1 of a neighboring subunit. A good control experiment would be, for example, to coexpress E241C subunits (no Trp) with H179W/H188W subunits (no Cys), to see whether the fluorescence signal is affected by interaction with Trp in nearby subunits.

One concern is that the effects of the lipid are entirely due to light scattering (excitation and emission), instead of interacting with the fluorophore. This concern is supported by the emission spectrum in the presence of PC, which seems to be the same emission peak as in 0 PC "sitting" on a shoulder (Fig.4A). An important control is to repeat the experiment in Fig.4A,B in the absence of fluorophore, to show whether any excitation light scattering affects the emission measurements.

The model in Fig.7 is simplistic and does not make complete sense: for example, in the H188W case (3rd row), at the intermediate state (center), the fluorophore is smaller (dimmer) than the depolarized condition (right), even though it is farther from the Trp quencher. A big advantage of the authors' work is understanding how the fluorophore interacts with the quenchers and its environment; Including an interpretative schematic is a great idea, and I think this one can be improved.

Some supplemental figures (2, 3 and 6) are not referenced in the main text. They should all be mentioned, but I believe the authors should discuss Supp. Fig. 6 in particular. What does it mean?

Finally, the 3-state model can account for the data well. An alternative (and similarly parsimonious) interpretation is that there are only two states, but the fluorophore is quenched by two distinct mechanisms, with unequal kinetics. Could the authors explore and confirm or exclude this interpretation?

Minor concerns:

Fig.1 is mistakenly cited as Fig.2 in the first Results paragraph. Likewise, a reference to Fig.1 (line 138) in fact refers to Fig.2. These may not be the only mistakes, please check carefully.

In general, the figures (also supplemental) should be numbered according to the order they are discussed in the text.

Please add citations for UCSF Chimera, ChimeraX, Jalview, and any other free academic software.

In Fig.7, shouldn't the third conformational state be "Active"? "Depolarized" describes the membrane potential, not the protein state. Fig.7 legend also has references to A, B, C and states #1 #2 #3, but none of these are in the figure.

Reviewer #2 (Remarks to the Author):

The manuscript by Papp et al. investigates the nature of the voltage-clamp fluorometry (VCF) signal using the voltage-gated proton channel Hv1. The VCF technique has been used to study conformational changes of various ion channels and other transmembrane proteins in the past decades. Fluorescence changes can come about by protein conformational changes that alter the immediate environment around the fluorophore, e.g. changes in the solvent properties or changes in the proximity of a quencher. While several important insights on kinetics and voltage dependence of conformational changes have been derived from these studies, the generation of the VCF signal, i.e. the molecular mechanism that leads to changes in the fluorescence intensity of the environmentally sensitive fluorophore, is often not well understood or left unstudied. The authors aim to fill this gap by investigating the nature of the VCF signal recorded on a particular labeling site at the extracellular loop between the S3 and S4 transmembrane segments, position 241, of the Ciona intestinales Hv1 channel. To this end, amino acids that might contribute to the VCF signal by collisional quenching are mutated and the phospholipid phosphatidylcholine is applied to make the extracellular solvent "more hydrophobic". The authors identify different components of the relatively complex VCF signal that they attribute to quenching and change in solvent exposure effects.

The study is a worthwhile and detailed investigation on the molecular mechanism of a VCF signal of Hv1, and the results will be helpful for ion- and proton-channel researchers. The methodology is overall sound and appropriate. However, there are several major points that need to be addressed to make the data, conclusions, and interpretations more convincing. The gain of knowledge about voltage sensing and gating of Hv1 is somewhat limited but also not the main focus of the study.

Major points:

1. The authors base their conclusions on only one particular labeling site on ciHv1. Several labeling sites on ciHv1 have been reported in previous studies. The authors should strengthen their experimental evidence and show Trp and/or lipid contribution to the VCF signal of at least one other labeling site. In addition, though not equally important, the authors could use iodide as a collisional quencher to provide more information about solvent-induced components in the VCF signal.
2. The authors interpret changes in VCF signal in Trp and Ala mutants as changes on the. However, introducing single point mutations can profoundly alter the biophysics of an ion channel. How can the authors exclude that the introduced mutations alter gating and therefore change the VCF signal?
3. Qiu et al. (Neuron 2013) used a labeling site adjacent to the one used in this study and reported that the fluorescence "hook" is nearly absent in monomerized Hv1 and putatively caused by dimer

interaction via an aspartate on transmembrane segment S1. How do the authors reconcile their data interpretation on quenching by amino-acid side chains with those of Qiu et al.? The authors should clarify this by testing the effect of Trp and His quenching in the monomerized channel.

4. Along the same line, the authors do not consider effects like self-quenching that might contribute to their VCF signal in a dimeric channel with two fluorophores in close proximity. Using concatamers it would be possible to label only one of the subunits exclusively so that such a mechanism (which in any case should be mentioned and discussed as a mechanism putatively contributing to the VCF signal) can be estimated.

5. Throughout the manuscript, the authors compare VCF signal amplitudes between different mutants, e.g. in Figure 3 between different Trp mutants, and interpret the differences in amplitude as a result of more efficient quenching. This interpretation is, however, problematic because changes in VCF signal intensity could also stem from different expression levels of the different mutants. Proper controls, e.g. detailed comparison of current amplitudes, kinetics, voltage dependence, fluorescence intensities etc., are required to make the data more compelling.

Minor points:

1. The authors should not name the ciHv1-E241C mutant as WT but rather name it ciHv1-E241C in text, legends, and figures.

2. p.2, lines 54-56, the statement that VCF is a combination of TEVC and SCAM is inaccurate. While the experimental purpose and the labeling strategy of VCF and SCAM are similar, the methods and readouts (fluorescence changes vs. binding kinetics) are very different.

3. p.2, the authors should define and consistently use "VCF" or "fluorescence" signal

4. p.3 line 93, the cited X-ray structure of mHv1 is not a dimer but a trimer, most likely because the structure is a chimeric protein contain a coiled coil that promotes trimerization.

5. p.3 line 95, citations missing.

6. p.3 line 110, De-La-Rosa & Ramsey Biophys J. 2020 should be also cited here.

7. p.4 line 126ff, Figures incorrectly referenced. The text reference to Figures 1 & 2 seems to be mixed up.

8. p.4 first paragraph, interpretation that there is no difference between the signals not convincing.

9. Figure 1E, the rationale of comparing different normalized VCF signals (Ftail vs Fhook) of the different mutants is not clear to me.

10. The recorded current traces should be displayed along with the voltage and fluorescence traces in the main figures.

11. Are the proton currents affected when PC is washed in? The authors should show the actual current traces (see comment above).

12. Suppl. Figs. 3 & 4 are not mentioned in the main text.

13. Figure 3, the VCF signal of H179W H188A is positive while the VCF signal of the other mutants is negative. What is the explanation for this, the proposed mechanism? How do model traces of H179W-H188A look like? The authors should include this in Fig. 6.

14. The authors did not report on the size of the samples they measured ("n"), so it is really hard to judge the validity of their data.

Reviewer #3 (Remarks to the Author):

The authors here propose a mechanism underlying the voltage-clamp fluorometry signals from the voltage-gated proton channel. They found that the signal is due to both quenching amino acids and the lipid environment around the fluorophore. Their 3-state model can explain the fluorometry signals from the wt and mutant Hv1 channels. Also, the model agrees with the previous model for Hv1 channel activation in terms of the two movements of the voltage sensor in Hv1 channels. The conclusions are convincing, but some of the analysis could be improved on. However, my main concern is that the conclusions are not especially novel and limitedly specific to Hv1 channels and

would not excite general readers.

Major comments.

1. The fluorescence signal from H179 H188A and H179W H188A looks very similar in shape. Does that mean that H179W is not interacting with the fluorophore? The increase in amplitude could just be due to increased expression?
2. Fig. 5. The subtraction is not a reliable method to separate the different quenching effects. How do you normalize the two recordings, since different oocytes and different mutations will have different expression levels? Small difference in expression levels will give large artifacts in the subtraction.
3. And quenching is not additive unless both quenching mechanisms are quenching the fluorophore very little. E.g. if either one of the quenchers quenches the signal by 90%, then both quenchers applied at the same time are not going to quench the fluorescence by 180%. The small fluorescence changes seen (<20%) is not an indication that the individual fluorophores are not quenched substantially, because background fluorescence contributes most likely a lot to the overall fluorescence signal.
4. Model. Why change charge and energy barriers for the different mutants? Do you think the mutants change the conformational changes that much?
5. In your cartoon it is mainly F3 that is affected by the mutants. So why not keep F1 and F2 the same for the different mutants, and just change F3?
6. At least try a simple model with just changing F3 and keep z , rates, F1, and F2 the same. If your cartoon is correct, then it should at least give a simulation that is qualitative similar to the data.
7. Please give the rate constants for all transitions in the model.

Minor comments.

1. Line 165. Tetramethylrhodamine Methyl Ester should be Tetramethylrhodamine Maleimide.
2. Suppl Fig. 2. What do all these signals mean? If they do not all have a meaning, then I would leave some of them out.

Reviewer #1 (Remarks to the Author):

In this work, Papp and colleagues investigate the activation mechanism of the voltage-gated proton channel Hv1 using the voltage-clamp fluorometry (VCF) approach. This is an experimental technique that allows one to track the conformational changes of a protein, as an attached fluorescent probe is differentially quenched at various protein states. The optical signal is sometimes composite of several quenching processes, making it difficult to interpret. Papp and colleagues investigate several different mechanisms that underlie the fluorescence signal, to understand better how the Hv1 channel can activate.

Several novel ideas arise from this work. The authors show how histidine residues can be fluorescence quenchers during VCF experiments and provide an experimental means to test the effect of lipids on fluorescence signals. An important advantage of their work is that it can provide a straightforward structural interpretation of the VCF findings. As more protein atomistic structures are increasingly made available by the cryo-EM revolution, VCF is one of the few approaches that can add a dynamic perspective to these structures during conformational changes pertinent to their function; so improving our ability to interpret VCF signals in terms of protein structure is highly valuable.

The work is of high experimental quality, but the conclusions require some additional analysis and control experiments. I believe they should be feasible for the authors.

Q1: First, the fluorescence signals are generally shown as the amplitude of fluorescence change over background fluorescence (DF/F). Oocytes can have a large variability of expression level, which would in turn affect the amplitude of the signals. Likewise, **background fluorescence can be affected by oocyte-to-oocyte variability of animal pole pigmentation (dark oocytes allow for less autofluorescence)**. A better way to quantify DF is normalization by maximal G , or (assuming the channel voltage dependence is not affected by the mutations) **current at a saturating membrane potential**. This will clarify whether DF are (for example) small because the fluorophore is quenched less, or there is less expression in that oocyte / channel clone, **or the oocyte had higher autofluorescence**. This is particularly important since DF amplitude is used in subtractions, and more advanced mathematical fitting.

We thank Reviewer #1 for the helpful comments and suggestions.

A1: The referees have criticized in multiple cases the comparison of VCF signal amplitudes and signal subtractions due to the many factors that can introduce large variability in these values. We agree that these points are justified and have accepted the fact that drawing conclusions from them is not well-founded. In the process of answering these points we have realized that

comparison of signal amplitudes across various mutants is not even necessary to support the main statements of our paper, it is the signal shapes that carry the information. We have therefore restricted our amplitude comparisons to self-control experiments where the same cell was exposed to control or lipid solutions, so the raised concerns can be avoided. Accordingly, all parts that discussed signal amplitude comparisons have been removed from the paper.

Nevertheless, we have performed the requested normalizations and answered the raised concerns, even though they are now not relevant for the main message of the paper.

Regarding the animal pole, we can confirm that the animal pole (dark half) of the oocyte emits less autofluorescence. This is why we always use the animal pole for recordings. It is also true that background fluorescence varies from oocyte to oocyte, sometimes independently from the expression level of the target-protein. We agree with that the normalization of dF/F by maximal G , or current at a saturating membrane potential could be a good attempt to eliminate the effect of different expression levels. However, we would like to note that all of this could only work well if the different mutations do not significantly affect channel conductance.

We have compared the whole cell currents of the various mutants and significantly smaller current was measured for all mutants compared to H179 H188 (one-way ANOVA).

Normalization of dF/F to the current measured at +100mV showed a significant difference for the double Ala mutation (H179A H188A) compared to H179 H188 (one-way ANOVA). Based on these results we have to conclude that the mutations can modify the expression level and/or the single channel conductance, therefore comparison of signal magnitudes is not justified.

Q2: The authors assume that the fluorescence signals arise from the interaction of the fluorophore with quenching groups in the same Hv1 subunit. But could they also arise from **interactions with residues in a neighboring subunit, either nearby or associated as a dimer?** For example, Mony et al (Nat Struct Mol Biol, 2015) showed that a fluorophore in S1 can be quenched by a Trp in S1 of a neighboring subunit. A good control experiment would be, for example, to **coexpress** E241C subunits (no Trp) with H179W/H188W subunits (**no Cys**), to see whether the fluorescence signal is affected by interaction with Trp in nearby subunits.

A2: We have performed the suggested measurement, mixing RNA of E241C subunits (no Trp) with H179W/H188W subunits (no Cys) in a 20%-80% ratio, respectively. In this way, in the majority of dimer Hv1 which can produce a VCF signal (89%, based on binomial distribution calculation), the TAMRA-labeled subunit was paired with a subunit which does not contain labelable Cys but two Trp residues. The resulting VCF signal (manuscript Fig. 8A) is **very similar** to the one measured for CiHv1-E241C (manuscript Fig. 5E). This leads to the conclusion that the dominant quenching residue must be located on the same subunit as the TAMRA dye. When lipid was added to the external solution during the mixed-RNA measurement, no significant change was observed in the magnitude of the VCF signal. This is in good agreement with what was measured for CiHv1-E241C with 2 His residues, supporting the idea that residues on the neighboring subunit do not influence the lipid effect.

We have repeated the same RNA-mixing experiment with the double Ala construct: 20% RNA of E241C subunits (no Trp) and 80% RNA of H179A/H188A subunits (no Cys). The measured VCF signals were almost identical to the double Trp mixed-RNA ones, further supporting our conclusion that the dominant quenching amino acids are located on the same subunit as the TAMRA dye. So, it doesn't matter whether Trp or Ala is on the other subunit, the VCF signal is the same. That is, the quenching amino acid is not on the other subunit, but on the same subunit as the TAMRA.

E H179 H188

A E241C (20%) + H179W H188W (80%)

B E241C (20%) + H179A H188A (80%)

Figure 7. Fluorescence traces of heterodimer CiHv1 and the effect of PC (phosphatidylcholine)

Representative fluorescence responses to a voltage step from -60 mV to +100 mV recorded from oocytes labeled with TAMRA-MTS, expressing heterodimer Ci-Hv1, mixing RNA of E241C subunits (Cys) with subunits (no Cys) in a 20%-80% ratio, respectively. (A) E241C subunits (Cys) with subunits (no Cys) H179W/H188W (N=5) (B), H179A/H188A (N=5). Left panels show the control VCF signals in the absence of PC, while the middle panels in the presence of 5mM PC in the extracellular solution. Right panel bar charts summarize the average PC effects. Vertical axis shows $\Delta F/F_h$ values, normalized to the control value (0mM PC). Error bars represent SEM.

Our results do not contradict the result published by the Isacoff lab in 2015 (doi:10.1038/nsmb.2978), where Trp on one subunit of the Hv1 tandem dimer was able to quench the fluorescence of TAMRA-MTS on the other subunit.

Figure 5. S1 undergoes voltage-dependent motion

(a) VCF fluorescence traces of wt-td-174C and V174W-td-174C Hv1 tandem dimers labeled with TAMRA-MTS (wt-td-174C* and V174W-td-174C*) for a step from -80 to $+100$ mV. Traces represent the average of 20 recordings on the same cell. (b) Magnitude of fluorescence changes of wt-td-174C* ($n = 7$ oocytes) and V174W-td-174C* ($n = 9$ oocytes). $p < 0.001$, two-tailed Student's t-test. Error bars, s.e.m.

It is very likely that a naturally dimerizing Hv1 is present in a different arrangement in the cell membrane from a tandem dimer, where the two subunits are bound together to form a single polypeptide chain.

According to results from the MacKinnon laboratory in 2008 (<https://doi.org/10.1073/pnas.0803277105>), in naturally dimerizing Hv1, the S1 helices are closest together and the S3-S4 loops are farthest apart. We have labeled the S3-S4 loop with TAMRA, so it is highly likely that it is not primarily the quenching amino acids of the other subunit but the quenching amino acids of the same subunit that play a role in the formation of the VCF signal.

Fig. 5. A model of the Hv1 dimer. (a) Topology diagram showing the dimer arrangement of Hv1 transmembrane regions based on the cross-linking data, viewed from the extracellular side. An ellipsoid denotes the 2-fold rotation axis normal to the membrane. (b) A cartoon representation of the Hv1 dimer. Transmembrane helices are labeled. The helical representation of the putative interfacial region (after S4) and the predicted coiled-coil region are also included.

Based on recent results from the Isacoff lab (2020, 10.1073/pnas.2010032117), we have constructed a dimer model of CiHv1 and determined the distances between positions E241 (where TAMRA is attached) and H179, H188. We obtained that the smallest distance is between position 188 on the same subunit. This is in agreement with our VCF measurement results, which indicate that the quenching amino acid at position 188 is the dominant one. In conclusion, all these new findings support our original hypothesis that fluorescence is dominantly quenched by the residues located on the same residue as TAMRA.

Q3: One concern is that the effects of the lipid are entirely due to light scattering (excitation and emission), instead of interacting with the fluorophore. This concern is supported by the emission spectrum in the presence of PC, which seems to be the same emission peak as in 0 PC “sitting” on a shoulder (Fig.4A). An important control is to repeat the experiment in Fig.4 A, B **in the absence of fluorophore** to show whether any excitation light scattering affects the emission measurements.

A3: The reviewer is absolutely right: both excitation and emission light are scattered by lipid molecules. We performed the control experiments that address this question at the very beginning.

We forgot to highlight that the results shown in figure 4, panel A are corrected for the results of TAMRA-free measurement. We have certainly not emphasized this enough. We subtracted the result of TAMRA-free measurement point by point from all the recordings with TAMRA. These figures above show spectrofluorimetry measurements of aqueous solutions without (top) and with TAMRA-MTS (middle), in a cuvette (excitation wavelength: 540 nm). The bottom one is the corrected one (subtracted the top from the middle one point by point).

Q4: The model in Fig.7 is simplistic and does not make complete sense: for example, in the H188W case (3rd row), at the intermediate state (center), the fluorophore is smaller (dimmer) than the depolarized condition (right), even though it is farther from the Trp quencher. A big advantage of the authors' work is understanding how the fluorophore interacts with the quenchers and its environment; Including an interpretative schematic is a great idea, and I think this one can be improved.

A4: We thank to the reviewer for identifying ways in which the model/schematic depicted in our model figure was unclear and could lead to misunderstanding. Following the suggestions, we

have revised the schematic to clarify several points. In our revised schematic, we depict how voltage-dependent movements of the arginine-containing S4 helix may modify the fluorescence of a reporter attached to the S3-S4 loop both by shifting the fluorophore to an environment with a different polarity, and also by changing the distance between the fluorophore and potential quenching residues such as H188. Thus, the fluorescence of a given state depends upon both the distance from the fluorophore to quenching residues and the polarity of the fluorophore environment (the latter would effectively mean that the distance from the fluorophore to lipid bilayer).

Based on our initial fitting results, in all three constructs, the intermediate state has a markedly lower fluorescence intensity than the resting state (see Supplementary Table 1 in the originally submitted manuscript). This would not occur if the fluorescence **only** depended on the distance between the fluorophore and position 188, but is consistent with the fluorophore residing in a more polar environment. In contrast, the fluorescence of the activated state increases (with respect to the resting state) in the H188A construct, decreases for the H188 construct, and is heavily reduced in the H188W construct (see Supplementary Table 1 in the originally submitted manuscript.), consistent with the fluorophore having much greater access to position H188 in the depolarized state.

The size of orange star in the figure and the fluorescence value in the table represent the intensity of a single fluorophore in a given state, while the F value calculated by the model represents the observed macroscopic fluorescence, which are not necessarily proportional to each other due to the different occupancy of the states. In the original manuscript version this was the case for H188W, where in state 2 we obtained a lower fluorescence for a single TAMRA than in state 3 based on our model parameters, but the macroscopically measured VCF signal is not biphasic (as in the other two cases) but only mono-phasic, since the transition from state 2 to state 3 is fast, i.e. channels spend little time in state 2. Since the relationship between the microscopic and macroscopic fluorescence is not straightforward we tried to better emphasize this in the text as well.

However, we have refined the code of fitting and the new result is from fitting multiple oocyte measurements for fluorescence, while simultaneously fitting multiple constructs for kinetics. The values in the new table come from this refined fitting procedure. Now the microscopic F (for a single TAMRA) and the macroscopic F (measured VCF signal) are more consistent with each other, and the size of the stars in the figure well represents the fit parameters shown in Supp. Table 1.

We hope this revised interpretive schematic is much clearer!

Q5: Some supplemental figures (2, 3 and 6) are not referenced in the main text. They should all be mentioned, but I believe the authors should discuss Supp. Fig. 6 in particular. What does it mean?

A5: The information shown on this Suppl. Fig. 6 can also be found in the table. This figure merely visualizes the free energy obtained from our model as a function of electric charge.

Since it is based on the values in the table and does not add any new information (it simply displays the values in the table in a bare visual form), we have removed this figure.

The references to Suppl. Fig. 2 and 3 have been added in the text.

Q6: Finally, the 3-state model can account for the data well. An alternative (and similarly parsimonious) interpretation is that there are only two states, but the fluorophore is quenched by two distinct mechanisms, with unequal kinetics. Could the authors explore and **confirm or exclude** this interpretation?

A6: The reviewer has raised an excellent point. Although one is usually most interested in the dynamics of the protein, Voltage-Clamp Fluorometry detects the combined dynamics of the protein and fluorophore. Thus, if the channel had only a single, voltage-dependent transition (e.g. R (Resting) \leftrightarrow A (Active)) but there were also fluorescence quenching mechanisms with millisecond kinetics (e.g. conformational change of the fluorophore itself, or a non-voltage dependent transition of the protein + fluorophore, between states 1 and 2), one could potentially observe multi-step fluorescence kinetics (e.g. R1 \rightarrow A1 \rightarrow A2 upon depolarization, and A2 \rightarrow R2 \rightarrow R1 upon repolarization).

However, in any model with a single voltage-dependent transition (i.e. 2 states for the channel), the voltage-dependence of the steady-state fluorescence and channel conductance must both follow the same single Boltzmann equation. This is not consistent with the experimental measurements (see Supp Fig 2), especially the H179 H188 (i.e. WT or CiHv1-E241C) and H179A-H188) constructs in which the steady-state fluorescence initially decreases for small depolarizations, before then increasing at larger depolarizations.

Thus, because the steady-state fluorescence has a non-monotonic dependence on voltage, this system cannot be described by a model with a single voltage-dependent transition. Instead, to describe the kinetics and voltage-dependence of the fluorescence signals any model must have at least two voltage-dependent transitions, and the 3-state model is the simplest possible example of such a model.

Minor concerns:

Q7: Fig.1 is mistakenly cited as Fig.2 in the first Results paragraph. Likewise, a reference to Fig.1 (line 138) in fact refers to Fig.2. These may not be the only mistakes, please check carefully.

A7: We have checked and corrected these errors.

Q8: In general, the figures (also supplemental) should be numbered according to the order they are discussed in the text.

A8: We have corrected this.

Q9: Please add citations for UCSF Chimera, ChimeraX, Jalview, and any other free academic software.

A9: We have added these citations.

Q10: In Fig.7, shouldn't the third conformational state be "**Active**"? "Depolarized" describes the membrane potential, not the protein state. Fig.7 legend also has references to A, B, C and states #1 #2 #3, but none of these are in the figure.

A10: We have modified this figure according to the suggestions.

Reviewer #2

Major points:

Q1: The authors base their conclusions on only one particular labeling site on ciHv1. Several labeling sites on ciHv1 have been reported in previous studies. The authors should strengthen their experimental evidence and **show Trp and/or lipid contribution to the VCF signal of at least one other labeling site.**

A1: We thank Reviewer #2 for this suggestion. We chose the S242 labeling site to perform the suggested experiments as this was successfully used previously by others for VCF. We made a construct, which contains the double Ala mutations (H179A, H188A) and a Cys residue at 242 position which was labeled with TAMRA. Using this construct, we performed the same VCF measurements and expected to see a significant lipid effect. Our results confirmed this expectation, lipids added to the extracellular solution significantly decreased the magnitude of the VCF signal amplitude (see below, 0.87 ± 0.04 , $p=0.01$, $N=6$).

When His residues are at positions 179 and 188, the lipid effect was not significant (see below, 0.92 ± 0.05 , $p=0.16$), which is also in good agreement with our results measured with E241C labeling site.

Using the double Trp construct (H179W, H188W) with S242C, we also measured a significant lipid effect (see below, 0.86 ± 0.02 , $p=0.001$, $N=7$), indicating that the lipid component is also important in the VCF signal formation. Amino acid quenching is still crucial, because the positive ΔF became negative for double Trp, indicating that the type of quenching amino acid is important at these critical positions (179,188). Also, we have to conclude that in this mutant (S242C/H179W/H188W), the TAMRA dye moves along a different pathway from when it is attached to E241C and along with the effect of the quenching of Trp residues, the lipid component remains detectable. This may seem contradictory at first glance (compared to the E241C, double Trp result), but we think that these mutations are likely to modify the basic structure of Hv1 and consequently the pathway of TAMRA during depolarization enough to explain this difference. We can still conclude that both the lipid effect and the quenching role of amino acids are crucial for the generation of the VCF signal, when labeling S242C with TAMRA, but the effect of Trp quenching is less dominant here than in the E241C mutant.

Q2: In addition, though not equally important, the authors could use **iodide as a collisional quencher to provide more information about solvent-induced components in the VCF signal.**

A2: We investigated the effect of collisional quenchers earlier (DOI: <https://doi.org/10.7554/eLife.42372.001>) on the TMEM266 protein. We tested several quenchers and found that KI at 50mM produced similar quenching as Trp amino acid at 15mM. In that work we used Trp in the extracellular solution as a collisional quencher and found that Trp produced a membrane potential dependent quenching (Fig. 4.).

We repeated this experiment now with CiHv1 S242C labeled with TAMRA. We obtained similar results: CiHv1 fluorescent signal was quenched by Trp dissolved in the extracellular solution and the quenching showed membrane potential dependence. But this membrane potential dependence was not a linear function (as it was for TMEM266) but seem to be correlated with the channel opening. These results clearly show that dissolved quenching molecules can interact with TAMRA attached to membrane proteins.

Q3: The authors interpret changes in VCF signal in Trp and Ala mutants as changes **on the**. However, introducing single point mutations can profoundly alter the biophysics of an ion channel. How can the authors exclude that the introduced mutations alter gating and therefore change the VCF signal?

A3: It is a reasonable assumption that even a single point mutation can completely change the gating and operation of an ion channel. We have thought about this and have therefore determined the G-V and F-V curves for each mutant channel. (Suppl. Fig. 2. and 3.). Suppl. Fig. 3D

and E summarize the G-V functions of the various constructs. We did not emphasize this enough in the original manuscript. These two panels show that there is a slight rightward shift due to both Ala and Trp mutations, but there is no fundamental change in the gating of the ion channel: the G-V function remained sigmoidal and the majority of ion channels are activated during the applied 1 sec long depolarizing pulse. We also have not observed any major changes in current kinetics.

Q4: Qiu et al. (Neuron 2013) used a labeling site adjacent to the one used in this study and reported that the fluorescence “hook” is nearly absent in monomerized Hv1 and putatively caused by dimer interaction via an aspartate on transmembrane segment S1. How do the authors reconcile their data interpretation on quenching by amino-acid side chains with those of Qiu et al.? The authors should clarify this by testing the effect of Trp and His quenching in the **monomerized channel**.

A4: We made the recommended constructs by inserting a stop codon at position 270. In these constructs the C-terminal part, the prolongation of S4, the coiled-coil region is missing which is necessary for dimerization. We made three such constructs: with double Ala mutations (H179A, H188A), with double Trp mutations (H179W, H188W) and with the original His residues (H179, H188) with E241C as a background mutation, which was labeled with TAMRA. Using these constructs, we performed VCF measurements. Our results at position 241 are similar to position 242 results published by Qiu et al., namely the hook disappeared in our VCF signals as well. We obtained monophasic, negative changes in fluorescence during depolarization with all the three monomer CiHv1 constructs (see below). In our interpretation, these results indicate that the TAMRA dye in monomer CiHv1 moves along a different pathway compared to the dimer. A monophasic decrease in fluorescence indicates that TAMRA moves away from the cell membrane and approaches the quenching residue at the same time. TAMRA could start to move from a deeper position, at a more embedded position into the cell membrane, then moves away from the lipid bilayer during depolarization and does not turn back toward the cell membrane. In this respect our results are in agreement with the results of Qiu et al. that the second conformational change is significantly affected in the monomeric channels due to the lack of inter-subunit interactions.

On the other hand, the fact that we were able to measure VCF signals with monomeric constructs supports the idea that the quencher amino acid(s) and TAMRA must be on the same subunit.

CiHv1 E241C V270Stop: Monomer & double His (N=6)

CiHv1 E241C V270Stop H179A H188A: Monomer & double Ala (N=8)

CiHv1 E241C V270Stop H179W H188W: Monomer & double Trp (N=7)

Q5: Along the same line, the authors do not consider effects like self-quenching that might contribute to their VCF signal in a dimeric channel with two fluorophores in close proximity. **Using concatamers** it would be possible to label only one of the subunits exclusively so that such a mechanism (which in any case should be mentioned and discussed as a mechanism putatively contributing to the VCF signal) can be estimated

A5: Reviewer #1 asked a related question where we reply that we mixed RNA of E241C subunits (no Trp) with H179W/H188W subunits (no Cys) in a 20%-80% ratio, respectively. In such way, in the majority of dimer Hv1 which can produce VCF signal (89%, based on binominal distribution calculation), the TAMRA-labeled subunit was paired with a subunit which does not contain labelable Cys but two Trp residues.

Using this construct, we labelled only one of the subunits exclusively as you recommended. The resulting VCF signal is very similar to the one measured for regular dimer CiHv1. We mixed the RNA of double Ala mutant construct as well which gave us almost identical VCF signal (see below).

Figure 7. Fluorescence traces of heterodimer CiHv1 and the effect of PC (phosphatidylcholine)

Representative fluorescence responses to a voltage step from -60 mV to +100 mV recorded from oocytes labeled with TAMRA-MTS, expressing heterodimer Ci-Hv1, mixing RNA of E241C subunits (Cys) with subunits (no Cys)H179W/H188W (N=5) (B), H179A/H188A (N=5). Left panels show the control VCF signals in the absence of PC, while the middle panels in the presence of 5mM PC in the

extracellular solution. Right panel bar charts summarize the average PC effects. Vertical axis shows $\Delta F/F_h$ values, normalized to the control value (0mM PC). Error bars represent SEM.

We think that, based on these results we can exclude the significant contribution of self-quenching between two fluorophores during the VCF signal generation.

Q6: Throughout the manuscript, the authors compare VCF signal amplitudes between different mutants, e.g. in Figure 3 between different Trp mutants, and interpret the differences in amplitude as a result of more efficient quenching. This interpretation is, however, problematic because changes in VCF signal intensity could also stem from different expression levels of the different mutants. **Proper controls, e.g. detailed comparison of current amplitudes, kinetics, voltage dependence, fluorescence intensities etc., are required to make the data more compelling.**

A6: We have received criticism in multiple cases related to the comparison of VCF signal amplitudes and signal subtractions due to the many factors that can introduce large variability in these values. We agree that these points are justified and have accepted the fact that drawing conclusions from them is not well-founded. In the process of answering these points we have realized that comparison of signal amplitudes across various mutants is not even necessary to support the main statements of our paper, it is the signal shapes that carry the information. We have therefore restricted our amplitude comparisons to self-control experiments where the same cell was exposed to control or lipid solutions, so the raised concerns can be avoided. Accordingly, all parts that discussed signal amplitude comparisons have been removed from the paper.

Nevertheless, we have performed the requested normalizations and answered the raised concerns, even though they are now not relevant for the main message of the paper.

As suggested by Reviewer #1, normalization of dF/F by maximal G , or current at a saturating membrane potential could be a good attempt to eliminate the effect of different expression levels. However, we would like to note that all of this could only work well if the different mutations do not significantly affect channel conductance.

The whole cell currents were compared, and significantly smaller current was measured for all mutants compared to H179 H188 using one-way ANOVA. Based on these results we have to conclude that the mutations can modify the expression level and/or the single channel conductance. But comparing the G - V functions measured for Ala and Trp mutations with the control G - V function (when there are His residues at positions 179 and 188), no significant differences can be observed. (Suppl. Fig. 3D and E)

Minor points:

Q7: The authors should not name the ciHv1-E241C mutant as WT but rather name it **ciHv1-E241C** in text, legends, and figures.

A7: We have corrected it according to the suggestion.

Q8: p.2, lines 54-56, the statement that VCF is a combination of TEVC and SCAM is **inaccurate**. While the experimental purpose and the labeling strategy of VCF and SCAM are similar, the methods and readouts (fluorescence changes vs. binding kinetics) are very different.

A8: We have rewritten this part.

Q9: p.2, the **authors should define** and consistently use “VCF” or “fluorescence” signal

A9: We have defined the VCF signal in the introduction:

„This fluorophore can then provide the VCF signal, by which we mean the change in the emitted fluorescence intensity divided by the holding fluorescence intensity, i.e. $\Delta F/F_h$ in response to conformational changes induced by membrane depolarization.”

Q10: p.3 line 93, the cited **X-ray structure of mHv1** is not a dimer but a **trimer**, most likely because the structure is a chimeric protein contain a coiled coil that promotes trimerization.

A10: We have corrected this statement.

Q11: p.3 line 95, **citations missing**.

A11: We have cited this paper:

Strong cooperativity between subunits in voltage-gated proton channels

Carlos Gonzalez, Hans P Koch, Ben M Drum & H Peter Larsson

Nature Structural & Molecular Biology volume 17, pages51–56 (2010)

Q12: p.3 line 110, **De-La-Rosa & Ramsey Biophys J. 2020 should be also cited** here.

A12: We have inserted this citation.

Q13: p.4 line 126ff, Figures incorrectly referenced. The text reference to Figures 1 & 2 seems to be mixed up.

A13: We have corrected this error.

Q14: p.4 first paragraph, interpretation that there is no difference between the signals not convincing.

A14: Due to the justified criticisms by the reviewers, all panels on the comparison of VCF signal amplitudes have been deleted from the manuscript. So now this question/comment is no longer relevant.

Q15: Figure 1E, the rationale of comparing different normalized VCF signals (F_{tail} vs F_{hook}) of the different mutants is not clear to me.

A15: We agree that the various evaluation methods were confounding, so we have changed the analysis process in the revision and compare only the ΔF_{tail} values.

Q16: The **recorded current traces should be displayed along with** the voltage and fluorescence traces in the main figures.

A16: In the original manuscript we had the current traces for each construct in Supplementary figures. However, at your request we have now added insets in Figs. 1 and 2 showing the measured proton currents.

Q17: Are the proton currents affected when PC is washed in? **The authors should show the actual current traces (see comment above).**

A17: This is a really relevant question. We performed this evaluation for all constructs where the lipid effect was tested. The lipid molecules applied extracellularly had no significant effect on the proton currents (see figure below). Furthermore, we put insets in Fig. 4 and 6 showing the measured proton currents as well.

Q18: Suppl. Figs. 3 & 4 are not mentioned in the main text.

We have inserted references to these figures in the appropriate places.

Q19: Figure 3, the VCF signal of H179W H188A is positive while the VCF signal of the other mutants is negative. What is the explanation for this, the proposed mechanism? **How do model traces of H179W-H188A look like?** The authors should include this in Fig. 6.

A19: As noted by the reviewer, during a strong depolarization (+100 mV) the VCF signal from the H179W H188A construct increases whereas for H179A H188W the fluorescence decreases. This would be very puzzling if the two positions were located right next to each other. However, residue H179 is located in the S1-S2 loop and in current structures of Hv1 is fully exposed to extracellular solution. In contrast, residue H188 is a good 1.5 turns into TM helix 2 and sits within the membrane. Since the two residues are physically separated and reside in different environments, the different impact of introducing a tryptophan at these two locations is not a total surprise.

Qualitatively, the fluorescence from both constructs is consistent with a mechanism in which the transition to the activated conformation increases the accessibility of the fluorescent reporter to the less polar environment close to position H188 (and thus stronger quenching from H188W) and decreases its accessibility to polar environment around position H179 (and thus less

quenching from H179W). This proposed mechanism is also quantitatively consistent with measured fluorescence. When both constructs are fitted to a common kinetic model (black lines),

the fluorescence intensity of the activated state is lower for H179A/H188W ($F_3 < F_1$) whereas it is higher for H179W/H188A ($F_3 > F_1$). However, it's important to note the mutations may also modify the channel kinetics as the fits improves if we allow for each construct to have different kinetics (grey lines).

Thus, the VCF signal of H179W H188A is consistent with the proposed model.

Q20: The authors did not report on the size of the samples they measured ("n"), so it is really hard to judge the validity of their data.

A20: In the manuscript, we modified the bar charts and put all the individual recordings on it and indicated the number of experiments (N=...).

Thank you for pointing this out.

Reviewer #3:

Major comments.

Q1: The fluorescence signal from H179 H188A and H179W H188A looks very similar in shape. Does that mean that H179W is not interacting with the fluorophore?

A1: Yes, we have arrived to the same conclusion, i.e. the quenching amino acid at position 188 has a more dominant quenching role in the VCF signal formation than the one at 179. This is supported by our distance measurements between the Cys amino acid at position 241 used for labeling and positions 179 and 188 on a structural model of CiHv1 (see below) based on available crystallographic data. We found that the amino acid at position 188 is closest to the amino acid at position 241 in the same subunit.

Q2: The increase in amplitude could just be due to increased expression?

A2: We have received criticism in multiple cases by all reviewers related to the comparison of VCF signal amplitudes and signal subtractions due to the many factors that can introduce large variability in these values. We agree that these points are justified and have accepted the fact that drawing conclusions from them is not well-founded. In the process of answering these points

we have realized that comparison of signal amplitudes across various mutants is not even necessary to support the main statements of our paper, it is the signal shapes that carry the information. We have therefore restricted our amplitude comparisons to self-control experiments where the same cell was exposed to control or lipid solutions, so the raised concerns can be avoided. Accordingly, all parts that discussed signal amplitude comparisons have been removed from the paper.

Nevertheless, we have performed the requested normalizations and answered the raised concerns, even though they are now not relevant for the main message of the paper.

The normalization of dF/F by maximal G , or current at a saturating membrane potential could be a good attempt to eliminate the effect of different expression levels. However, we would like to note that all of this could only work well if the different mutations do not significantly affect channel conductance.

The whole cell currents were compared, and significantly smaller current was measured for all mutants compared to H179 H188 using one-way ANOVA. Based on these results we have to conclude that the mutations can modify the expression level and/or the single channel conductance.

Q3: Fig. 5. The subtraction is not a reliable method to separate the different quenching effects. How do you normalize the two recordings, since different oocytes and different mutations will have different expression levels? Small difference in expression levels will give large artifacts in the subtraction.

A3: Please see A2. Accordingly, we have deleted this figure from this manuscript.

Q4: And quenching is not additive unless both quenching mechanisms are quenching the fluorophore very little. E.g. if either one of the quenchers quenches the signal by 90%, then both quenchers applied at the same time are not going to quench the fluorescence by 180%.

A4: This is obviously true. We think this should be calculated so that if both are 90% quenched, then $0.1 \cdot 0.1 = 0.01$ of the signal remains as in the case of two independent channel blockers.

But since we deleted the referenced figure here, this issue will no longer be relevant.

Q5: The small fluorescence changes seen (<20%) is not an indication that the individual fluorophores are not quenched substantially, because background fluorescence **contributes most likely a lot** to the overall fluorescence signal.

A5: The magnitude of the VCF signal can indeed be affected, but not its shape. Since the parts of the figures about magnitude have been deleted, this question/comment is no longer relevant.

Q6: Model. Why change charge and energy barriers for the different mutants? Do you think the mutants change the conformational changes that much?

A6: Since mutations often alter reaction kinetics, we deliberately allowed for this possibility in our original analysis. However, the reviewer makes an excellent point that these mutations could primarily influence the fluorescence intensity of the conformations, and not measurably change the reaction kinetics.

To test the reviewer's idea we have performed additional analysis described below.

Q7: In your cartoon it is mainly F3 that is affected by the mutants. So why not keep F1 and F2 the same for the different mutants, and just change F3?

A7: This a good idea to test, and we have effectively implemented it in the additional analysis as seen below.

Q8: At least try a simple model with just changing F3 and keep z , rates, F1, and F2 the same. If your cartoon is correct, then it should at least give a simulation that is qualitative similar to the data.

A8: This is an excellent suggestion, we have tested it. The results are shown below.

The black lines correspond to a fit in which mutations only alter the fluorescence of the final activated state, F_3 , (i.e. all constructs have the same energetic landscape (i.e. charge and free energy of the states and transition intermediates) and thus reaction kinetics).

The grey lines represent a fit in which each construct can have a unique energy landscape (i.e. mutations are permitted to change the charge and free energy of the states and transition intermediates and thus reaction kinetics).

Although the grey lines are clearly a better fit, as the reviewer suggested "the simple model with just changing F_3 and keep z , rates, F_1 , and F_2 the same" is indeed qualitatively similar to the data.

Furthermore, the fitted magnitude of F3 (relative to F2) also agrees with the cartoon (i.e. strongest quenching for H188W, some quenching for 179H/188H and 179A/188H, and miniscule dequenching for H188A).

Thus, we hope the reviewer agrees with us that the model depicted in the cartoon is largely consistent with the experimental data.

Q9: Please give the rate constants for all transitions in the model.

A9: The rate constants can be readily calculated from the charge and free energy of the states and transition intermediates, but we appreciate the reviewers suggestion to explicitly present the transition rates. Because the rates are voltage dependent, we think it might be most helpful to depict them graphically, along with the steady-state occupancy of the three states:

y

Minor comments.

Q10: Line 165. Tetramethylrhodamine Methyl Ester should be Tetramethylrhodamine Maleimide.

A10: We have corrected it as suggested.

Q11: Suppl Fig. 2. What do all these signals mean? If they do not all have a meaning, then I would leave some of them out.

In the original manuscript we showed F-V curves calculated for various components of the complex VCF signal (e.g. hook, tail, etc.) along with the G-V function for each construct. But we agree that they do not add useful information to the understanding of the manuscript. We have therefore removed most of them leaving only the F-V calculated from the tail amplitude along with the G-V function for each construct. Since it is reasonable to assume that even a single point mutation can change the gating and operation of the channel we have compared the G-V and F-V curves of the channels (Suppl. Fig. 2. and 3.). Suppl. Fig. 3D and E summarize the G-V functions of the various constructs. This was not sufficiently emphasized in the original version. These two panels show that there is a slight rightward shift due to both Ala and Trp mutations, but there is no fundamental change in the gating of the ion channel: the G-V function remained sigmoidal and the majority of ion channels are activated during the applied 1 sec long depolarizing pulse.

Reviewers' comments:

Reviewer #1 (Remarks to the Author):

Papp and colleagues have substantially revised their work in response to my comments.

There is one point that I felt was not adequately addressed and, as it is, I believe that it impacts the quality of the work, rather than its conclusions.

It has to do with Q3, the experiment of TAMRA with increasing lipid in the spectrofluorometer. I thank the authors for clarifying that the figure 4A (5A in the revised version) is showing subtracted spectra (TAMRA+PC - PC alone), and for sharing the raw spectra in the rebuttal. Maybe I am missing something, but I am not convinced that the subtraction worked well. Note that the orange trace in 5A (5mM PC) shows substantial "emission" at 550 nm, in contrast to the grey (0mM PC) trace. Second, in the 5mM trace, going from 550nm to higher wavelengths, the emission decreases (as does the shoulder in the PC-spectra alone); in the 0mM trace, the emission increases, as expected from the TAMRA solution.

Looking at the raw traces, it looks like [PC] is overestimated in the PC-alone spectra. I believe this because of the following observations: 1) in the 1uM TAMRA, 0mM PC condition, $F@550$ is ca. $0.4e5$. 2) In 0uM TAMRA, 5mM PC, $F@550=2e5$. 3) The authors' analysis (Fig.5B) shows that TAMRA emission increases by up to 2-fold in 5mM PC; but even in this extreme case, one expects that in the 1uM TAMRA, 5mM PC condition, $F@550$ would be $2 \times 0.4e5 + 2e5 = 2.8e5$, but in fact it is $3.5e5$.

Based on these observations, my interpretation of the orange trace in 5A is that the emission of TAMRA is in fact less than in 0 mM PC, but it sits on a shoulder of insufficiently-subtracted scattered light. As is, I don't think that the authors can claim that "the emission intensity increased in a concentration dependent manner" (p.5), unless they can consistently eliminate this shoulder in their subtracted spectra. The authors have probably considered this already, but one practical idea to ensure consistent [PC] and [TAMRA] is by splitting and mixing pre-diluted PC and TAMRA solutions, rather than making new dilutions from scratch for each measurement.

If subtracting the shoulder is not practically possible, my suggestion is to concede that this experiment cannot provide a clear enough measurement and exclude it from this work. Since the results in VCF show that PC does not preferentially affect a specific fluorescence component, I do not believe that this experiment is necessary for the authors' conclusions.

Reviewer #2 (Remarks to the Author):

The authors addressed my primary concern by showing lipid quenching data on an additional labeling site, S242C. While I appreciate the authors' efforts, I am still concerned about the interpretation of the fluorescence data. The relative change in the fluorescence signal, which is quite modest in all shown mutants, could be a rundown (or bleaching) effect rather than quenching by lipids (Rundown and bleaching is common in VCF data). The authors should provide convincing evidence that this is indeed no rundown, e.g. by showing that the amplitude of the fluorescence signal increases after lipid washout. If wash-out of lipids is problematic, the authors could show how the time course of the lipid wash-in, i.e. continuously repeat stimulations while washing in lipids, and compare this with control wash-in of standard solution. Like this, the authors can convince the reader that the lipid effect is specific. In addition, the authors should provide these data in the manuscript, not just in the reply. This point is important because the author's claim that added lipids quench fluorescence signals of ciHv1-H179A-H188A is the sole strong evidence overall and the major novelty of the manuscript (trp quenching, as mentioned by the first reviewer, has been shown before).

Reviewer #3 (Remarks to the Author):

The authors have responded well to my comments.

Reviewer #1 (Remarks to the Author):

Papp and colleagues have substantially revised their work in response to my comments.

There is one point that I felt was not adequately addressed and, as it is, I believe that it impacts the quality of the work, rather than its conclusions.

It has to do with Q3, the experiment of TAMRA with increasing lipid in the spectrofluorometer. I thank the authors for clarifying that the figure 4A (5A in the revised version) is showing subtracted spectra (TAMRA+PC - PC alone), and for sharing the raw spectra in the rebuttal. Maybe I am missing something, but I am not convinced that the subtraction worked well. Note that the orange trace in 5A (5mM PC) shows substantial "emission" at 550 nm, in contrast to the grey (0mM PC) trace. Second, in the 5mM trace, going from 550nm to higher wavelengths, the emission decreases (as does the shoulder in the PC-spectra alone); in the 0mM trace, the emission increases, as expected from the TAMRA solution.

Looking at the raw traces, it looks like [PC] is overestimated in the PC-alone spectra. I believe this because of the following observations: 1) in the 1uM TAMRA, 0mM PC condition, $F@550$ is ca. $0.4e5$. 2) In 0uM TAMRA, 5mM PC, $F@550=2e5$. 3) The authors' analysis (Fig.5B) shows that TAMRA emission increases by up to 2-fold in 5mM PC; but even in this extreme case, one expects that in the 1uM TAMRA, 5mM PC condition, $F@550$ would be $2 \times 0.4e5 + 2e5 = 2.8e5$, but in fact it is $3.5e5$.

Based on these observations, my interpretation of the orange trace in 5A is that the emission of TAMRA is in fact less than in 0 mM PC, but it sits on a shoulder of insufficiently-subtracted scattered light. As is, I don't think that the authors can claim that "the emission intensity increased in a concentration dependent manner" (p.5), unless they can consistently eliminate this shoulder in their subtracted spectra. The authors have probably considered this already, but one practical idea to ensure consistent [PC] and [TAMRA] is by splitting and mixing pre-diluted PC and TAMRA solutions, rather than making new dilutions from scratch for each measurement.

If subtracting the shoulder is not practically possible, my suggestion is to concede that this experiment cannot provide a clear enough measurement and exclude it from this work. Since the results in VCF show that PC does not preferentially affect a specific fluorescence component, I do not believe that this experiment is necessary for the authors' conclusions.

We thank Reviewer #1 for pointing out the “shoulder” in the spectrofluorimetric measurement results, which indicates that there was an error either in the measurement or in the correction calculation process. We therefore repeated this measurement entirely, taking great care to calculate and measure the lipid concentrations.

Although the new measurements yielded slightly higher intensities, the shape of the emission spectrum and the position of the maximum remained unchanged. The excitation wavelength was changed from 540 nm to 535 nm, as it was considered possible that erroneous correction of the excitation light scattering contributed to the appearance of the shoulder part of the spectrum.

We repeated the measurements 4 times and the shoulder disappeared completely. See figure below.

The new set of experiments reproduced the previous results: we observed an increase in fluorescence intensity with increasing lipid concentration, but a significant increase was still only measured at 5 mM lipid concentration.

Reviewer #2

The authors addressed my primary concern by showing lipid quenching data on an additional labeling site, S242C. While I appreciate the authors' efforts, I am still concerned about the interpretation of the fluorescence data. The relative change in the fluorescence signal, which is quite modest in all shown mutants, could be a rundown (or bleaching) effect rather than quenching by lipids (Rundown and bleaching is common in VCF data). The authors should provide convincing evidence that this is indeed no rundown, e.g. by showing that the amplitude of the fluorescence signal increases after lipid washout. If wash-out of lipids is problematic, the authors could show how the time course of the lipid wash-in, i.e. continuously repeat stimulations while washing in lipids, and compare this with control wash-in of standard soluton. Like this, the authors can convince the reader that the lipid effect is specific. In addition, the authors should provide these data in the manuscript, not just

in the reply.

This point is important because the author's claim that added lipids quench fluorescence signals of ciHv1-H179A-H188A is the sole strong evidence overall and the major novelty of the manuscript (trp quenching, as mentioned by the first reviewer, has been shown before).

We thank Reviewer #2 for drawing our attention to the possibility that the decrease in VCF signal magnitude could be due to bleaching of the TAMRA or simply due to rundown phenomenon. Fortunately, we performed the original measurements by washing the lipid out from the external solution (with a solution containing 0mM lipid) after applying the increasing lipid concentrations.

Unfortunately, we did not show the washout results in the original figure, but now we have added a new column to the bar chart in Figure 5 for each of the three mutants to represent the lipid washout (also see it below), and agree that it is a valuable addition, which carries important information.

C H179A H188A

D H179W H188W

E H179 H188

REVIEWERS' COMMENTS:

Reviewer #1 (Remarks to the Author):

The authors have fully satisfied all my concerns.

Thank you for performing the extra experiments: I agree, the results with 535nm light are a lot more convincing.

Reviewer #2 (Remarks to the Author):

The authors addressed my concern about the wash-out in Figure 5 by placing bar graphs. They should also add raw fluorescence traces of the wash-out condition next to the 5 mM PC trace in Figure 5C-E for each of the three mutants. This is of critical importance, in particular for the H179A-H188A mutant.